# RLSF-V: Mitigating Hallucinations in MLLMs via Fuzzy Semantic Self-Feedback

**Changhao He**[1] **Shuhao Yan**[1] **Shuxian Li**[1] **Xi Peng**[2 3] **Peng Hu**[† 1]

 https://github.com/XLearning-SCU/RLSF-V

## Abstract

Multimodal large language models (MLLMs) extend large language models (LLMs) with visual perception for open-world understanding, but exacerbate LLMs' hallucinations, in which generated text contradicts visual evidence or common sense. To mitigate hallucinations, a dominant strategy is Direct Preference Optimization (DPO) using hallucination-labeled responses. Existing pipelines, however, face two key limitations: they either (i) rely on human inspection or proprietary models to correct hallucinated outputs, producing off-policy preference data that violate the assumptions of DPO, or (ii) depend on stronger models to evaluate responses, leading to an unfavorable trade-off between performance and scalability. Departing from these paradigms, we propose a reference-policy *self-feedback* framework that constructs preference data for hallucination mitigation without any external supervision (*e.g.*, large models or humans). Specifically, we present a novel *local fuzzy semantic* evaluation paradigm that derives a hallucination-sensitive confidence signal directly from the internal logits, which is then used to automatically rank diverse generated responses to build preference pairs for fine-tuning. Trained on a 10k-scale dataset, our method achieves competitive performance on both generative and discriminative benchmarks compared to existing RLHF and RLAIF baselines.

## 1. Introduction

Multimodal large language models (MLLMs), which jointly process visual and textual information, have achieved strong

[1]College of Computer Science, Sichuan University, Chengdu, China [2]School of Artificial Intelligence, Sichuan University, Chengdu, China [3]Tianfu Jincheng Laboratory, Chengdu, China. Correspondence to: Peng Hu <penghu.ml@gmail.com>.

*Proceedings of the 43rd International Conference on Machine Learning*, Seoul, South Korea. PMLR 306, 2026. Copyright 2026 by the author(s).

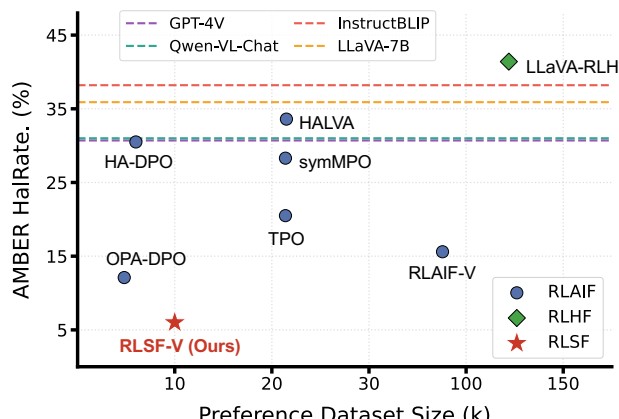

*Figure 1.* Hallucination rate *vs.* preference data size on AMBER. With only 10k-scale self-generated preference pairs, **RLSF-V** attains the lowest hallucination rate, outperforming both **RLHF** and **RLAIF** baselines while using a relatively small amount of preference data and no external large models.

performance across a wide range of tasks and attracted widespread attention in both research and industry (Feng et al., 2023b; Liu et al., 2024b; Li et al., 2025; Su et al., 2026; Qin et al., 2026; He et al., 2026; 2024a). However, due to the vision tower and the LLM component being usually trained separately and hard to align (Tong et al., 2024), MLLMs remain highly prone to hallucinations, leading models to describe nonexistent objects, assign incorrect attributes, or fabricate facts that do not match reality (Bai et al., 2024; Liu et al., 2024c). Such unfounded content can seriously undermine user trust and the practical utility of MLLMs, especially in domains where reliability and safety are critical (Feng et al., 2023a), such as medical question answering (Xu et al., 2024), embodied intelligence (Feng et al., 2026a;b; 2025), and autonomous driving (Cui et al., 2024; Jiang et al., 2025).

To mitigate hallucinations, prior work has generally followed two main directions based on whether additional training is required: (i) *training-free* methods that intervene in the decoding process at inference time, and (ii) *training-based* methods that explicitly align MLLMs toward non-hallucinatory behavior. Representative training-free approaches, such as contrastive decoding and its vari-

ants (Leng et al., 2024; Wang et al., 2024b; Huo et al., 2025), generate perturbed or degraded versions of the input and compare the model's responses to those for the original input. By contrasting these responses and filtering out patterns that are unstable under perturbations, such methods can effectively improve robustness to hallucinations (Zou et al., 2024). Training-based approaches, in contrast, design elaborate data-construction pipelines to obtain high-quality preference datasets, often involving manual annotation (Yu et al., 2024; Sun et al., 2024), response rewriting with advanced proprietary models (*e.g.*, GPT-4 (Achiam et al., 2023)), or evaluation by stronger models (Yu et al., 2025a;b). The resulting preference pairs (hallucinated *vs.* non-hallucinated) are then used to fine-tune MLLMs, typically via Direct Preference Optimization (DPO) (Rafailov et al., 2023), encouraging the model to prefer factually grounded outputs.

Despite their effectiveness, both lines of work suffer from notable limitations. First, decoding-based methods typically require multiple forward propagations per inference to capture different hallucination patterns (Park et al., 2025), which inevitably increases computational overhead and response latency (Yin et al., 2025a;b). Second, the performance of preference learning methods is highly sensitive to the quality of the preference data (Ouali et al., 2024; Xiao et al., 2025), often heavily relying on strong external evaluators to detect or correct hallucinations (Xie et al., 2024). This dependence makes such pipelines difficult to scale in practice (Yu et al., 2025b) and may raise privacy concerns (Xiao et al., 2024). Moreover, recent studies (Wu et al., 2025; Peng et al., 2025) have pointed out that preferences obtained via such external corrections are often *off-policy* (Schulman et al., 2017), as they are generated by independent evaluators whose output distributions differ from that of the reference model. This distributional mismatch violates the on-policy assumption underlying DPO objectives, which expect preference data to be collected on (or close to) the model's own behavior (Rafailov et al., 2023), and can therefore lead to suboptimal or even misaligned updates. Taken together, these limitations hinder the integration of existing hallucination-mitigation techniques into lightweight, reusable MLLM training and post-training workflows.

To address these challenges, we propose a simple yet effective self-feedback framework for hallucination mitigation that requires neither external large models nor human annotations. Concretely, for each multimodal prompt, we first generate multiple diverse candidate responses under identical conditions. We then present a local fuzzy semantic evaluation paradigm that derives a hallucination-sensitive confidence signal from the model's internal logits. This paradigm interprets the competition among top-ranked candidates as fuzzy membership and computes a sentence-level fuzzy semantic score for each answer. To better align this score

with hallucination behavior, we apply a lightweight part-of-speech (POS) tagger to identify content-bearing tokens (*e.g.*, NOUNS, NUMERALS, ADJECTIVES, VERBS, ADVERBS, *etc.*) and restrict the fuzzy semantic aggregation to these positions, which are most likely to carry object, attribute, and relation hallucinations. The responses with the lowest and highest resulting uncertainty are treated as chosen and rejected, respectively, yielding a reference-policy pairwise preference dataset. Finally, we use these preference pairs for MLLM fine-tuning, enabling hallucination mitigation without any external evaluators, response rewriting, or human supervision.

Our key contributions can be summarized as follows:

- We propose a self-feedback framework for constructing preference data that eliminates the need for external large-model evaluators or human annotations, and can be easily applied to diverse MLLMs.

- We present a novel hallucination assessment method that derives local fuzzy semantics directly from internal logits and seamlessly integrates into a preference optimization pipeline. We empirically show that this method outperforms classical evaluation baselines such as Shannon entropy or evidence theory.

- We reveal that hallucinations are primarily driven by a subset of content-bearing tokens and propose a POS-aware token-selection strategy that restricts our fuzzy semantic aggregation to the most informative positions, significantly improving hallucination evaluation performance.

- Extensive experiments against state-of-the-art hallucination mitigation methods and uncertainty quantification baselines demonstrate that RLSF-V achieves competitive performance without any external large models, while striking a favorable balance between scalability and effectiveness.

## 2. Related Work

### 2.1. Hallucination Mitigation in MLLMs

Hallucination mitigation in MLLMs is commonly addressed by either *training-free* decoding strategies or *preference-based* fine-tuning (Bai et al., 2024; Liu et al., 2024c; Lan et al., 2024). Decoding-time methods operate without updating model parameters, for example by perturbing visual (Leng et al., 2024) or textual inputs (Wang et al., 2024b), pruning visual tokens (Huo et al., 2025), or constructing contrastive prompts to expose and suppress hallucination patterns during inference (Wang et al.; Su et al., 2025; Kim et al., 2025; Chen et al., 2025b). These approaches typically assume that hallucination behaviors are stable across

different input degradations and usually require multiple forward passes per query, which can be costly and less suitable for on-device or low-latency deployment (Yin et al., 2025b). In addition, activation-steering methods (Su et al., 2025; Wang et al., 2025a) suppress undesired behaviors by modifying intermediate activations with pre-computed direction vectors, but their effectiveness typically depends on intervention-specific choices such as the construction of the steering direction, the target layer, the token position, and the steering strength. Preference-based methods (Wu et al., 2025; Fu et al.) instead fine-tune the model to favor non-hallucinatory responses over hallucinated ones by constructing explicit pairs of "good" versus "bad" outputs and optimizing a preference objective (*e.g.*, via DPO (Rafailov et al., 2023)). Such approaches amortize the cost of hallucination detection into training and can improve robustness at test time with a single decoding pass (Wang et al., 2024a; Xie et al., 2024). Our work follows this preference-learning paradigm rather than a decoding-based one.

## 2.2. Preference Learning from Feedback

Existing preference-learning pipelines for hallucination mitigation mainly differ in how preference data are obtained. RLHF-style methods (Sun et al., 2024) collect explicit human judgments on hallucinations and use them to build preference pairs (Yu et al., 2025c), while RLAIF-style methods (Liu et al., 2025; Yu et al., 2025b) rely on stronger external large models to score or rewrite responses (Xiao et al., 2025), treating model-validated or rewritten outputs as preferred. Although effective, both lines of work depend on either large-scale human annotation or continuous access to powerful external models. Furthermore, when preferred responses are heavily edited by humans or external models, the resulting preference data can become *off-policy* (Schulman et al., 2017; Liu et al., 2025; Rafailov et al., 2023) with respect to the reference model. Hybrid schemes that combine supervised fine-tuning with preference learning have been proposed to narrow this distribution gap (Wu et al., 2025), but they still require external supervision. In contrast, we propose a self-feedback framework that constructs preference data without any human labels or external large models. By introducing a hallucination-sensitive fuzzy semantics evaluation paradigm derived solely from the internal logits of the target MLLM, we rank diverse candidate responses and obtain high-/low-quality pairs in a fully self-contained manner, making them directly compatible with preference learning.

## 2.3. Uncertainty Quantification from Internal Signals

A standard way to extract confidence from internal signals is to compute token-level Shannon entropy (Kadavath et al., 2022) over the softmax distribution and aggregate it across positions (*e.g.*, by averaging). This relies on nor-malized probabilities over the full vocabulary and implicitly treats all tokens as equally informative (Chen et al., 2024a). In practice, the entropy can be dominated by the long tail of low-probability tokens, and uniform aggregation ignores that only a small subset of content-bearing tokens typically determines whether a response is hallucinatory (Duan et al., 2024). To go beyond plain entropy, prior work has explored evidential methods (Ma et al., 2025a) that jointly model aleatoric and epistemic uncertainty (*e.g.*, via Dirichlet evidence (Sensoy et al., 2018)), as well as energy-based scores (Ma et al., 2025b) inspired by Boltzmann distributions. While effective for classification and out-of-distribution detection, these formalisms still operate on probability simplices or global sequence scores and do not directly align with sentence-level hallucination risk in autoregressive generation (Chen et al., 2025a). In contrast, we propose a *fuzzy semantic* view of internal logits, modeling the competition among top candidates as fuzzy membership. By integrating the derived uncertainty signal with POS-aware token selection, our method effectively captures both the varying semantic importance of tokens and the local ambiguity associated with hallucinations.

## 3. Method

### 3.1. Diverse Candidate Generation

Let $p_\Theta$ denote the base MLLM with parameters $\Theta$. Given a multimodal prompt consisting of an image–text pair $(m, x)$, the model autoregressively generates a token sequence:

$$y^{(i)} = (y_1^{(i)}, \ldots, y_{T_i}^{(i)}), \quad y_t^{(i)} \in \mathcal{V}, \tag{1}$$

where $\mathcal{V}$ is the vocabulary, $T_i$ is the length of the $i$-th response, and the conditional distribution at each decoding step $t$ is:

$$p_\Theta(y_t^{(i)} \mid y_{<t}^{(i)}, m, x) = \frac{\exp(\mathbf{z}_{t, y_t^{(i)}}^{(i)})}{\sum_{v \in \mathcal{V}} \exp(\mathbf{z}_{t,v}^{(i)})}, \tag{2}$$

where $\mathbf{z}_t^{(i)} \in \mathbb{R}^{|\mathcal{V}|}$ denotes the output logit vector. To obtain a diverse set of candidates for each prompt, we sample $N$ responses from $p_\Theta(\cdot \mid m, x)$ using stochastic decoding strategies, including temperature, top-k, and nucleus sampling (Holtzman et al., 2020). All logits produced during decoding are cached and later used for uncertainty quantification, ensuring that uncertainty quantification is performed without querying any external models.

### 3.2. Local Fuzzy Semantics Aggregation

A popular way to quantify the model's uncertainty about a generated sequence $y^{(i)}$ is to compute token-level predictive Shannon entropy (Kadavath et al., 2022) and then aggregate it across positions. Specifically, for a fixed multimodal

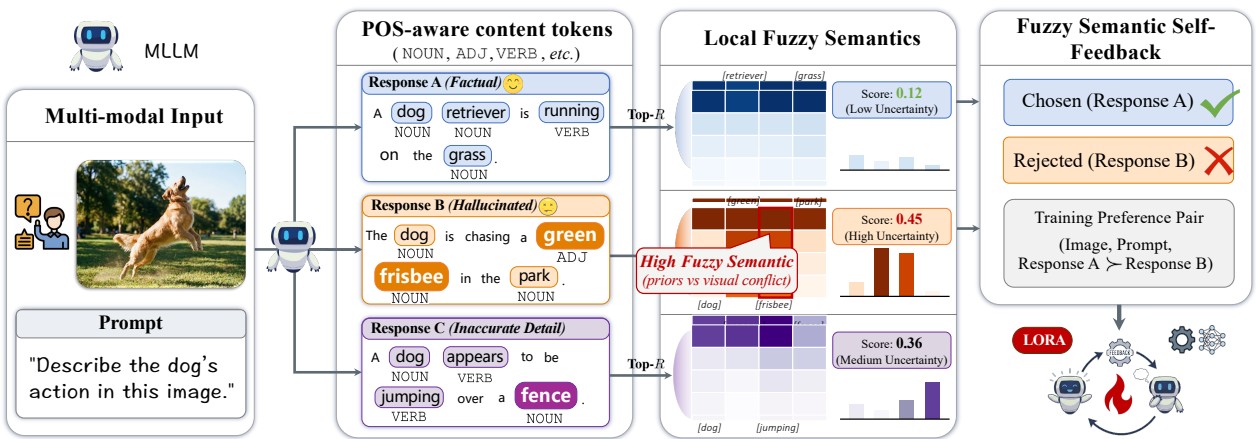

*Figure 2.* Overview of the proposed fuzzy semantic self-feedback pipeline (RLSF-V). Given a multi-modal input $(m, x)$, a base MLLM samples diverse candidate responses $\{y^{(i)}\}_{i=1}^{N}$. We apply token filtering and compute local fuzzy semantics from the model's internal logits to obtain a POS-aware fuzzy semantics score $U_{pos}(y^{(i)})$ for each candidate. The lowest-uncertainty response is selected as `chosen` and the highest-uncertainty response as `rejected`, forming self-generated preference pairs for MLLM fine-tuning.

prompt $(m, x)$ and response $y^{(i)}$, the predictive Shannon entropy at decoding step $t$ is:

$$H_t^{se} = -\sum_{v \in \mathcal{V}} p_\Theta(v \mid y_{<t}^{(i)}, m, x) \log p_\Theta(v \mid y_{<t}^{(i)}, m, x).$$

(3)

However, predictive Shannon entropy $H_t^{se}$ is defined on the softmax probabilities over the full vocabulary and is highly sensitive to the *multiplicative* scale of the logits. Specifically, as the logits are globally rescaled (equivalently, as the softmax temperature changes), $H_t^{se}$ can quickly saturate toward zero even when the relative margins between plausible candidates remain unchanged. As a result, the aggregated Shannon entropy score primarily reflects probability calibration rather than local ambiguity between competing continuations, and we find it poorly aligned with hallucination severity in MLLMs. To better capture the model's local indecision among plausible continuations, we reinterpret the output logits as a fuzzy membership function over candidate tokens:

**Definition 3.1 (Fuzzy membership and local fuzzy semantics).** For each position $t$, let $\mathcal{S}_t = \{v \in \mathcal{V} \mid z_{t,v} > -\infty\}$ denote the set of candidates with finite logits, and write $\{z_{t,v}\}_{v \in \mathcal{S}_t}$ for their logits. We map them into the unit interval via min–max normalization:

$$\mu_{t,v} = \frac{z_{t,v} - z_{t,\min}}{z_{t,\max} - z_{t,\min} + \epsilon}, \quad v \in \mathcal{S}_t,$$

(4)

where $z_{t,\max} = \max_{u \in \mathcal{S}_t} z_{t,u}$, $z_{t,\min} = \min_{u \in \mathcal{S}_t} z_{t,u}$, and $\epsilon$ is a small constant to avoid division by zero. The vector $\boldsymbol{\mu}_t = (\mu_{t,v})_{v \in \mathcal{S}_t}$ can be viewed as a fuzzy membership profile describing how strongly the model leans towards each candidate at step $t$, where values close to 1 indicate

strong support, while values near $0.5$ indicate ambiguity between competing options. Based on $\boldsymbol{\mu}_t$, the local fuzzy semantics (De Luca & Termini, 1993) at position $t$ is then defined as:

$$H_t^f = -\frac{1}{|\mathcal{S}_t| \ln 2} \sum_{v \in \mathcal{S}_t} \Big[ \mu_{t,v} \ln \mu_{t,v} + (1 - \mu_{t,v}) \ln(1 - \mu_{t,v}) \Big],$$

(5)

so that $H_t^f \in [0, 1]$.

In the context of token-level uncertainty quantification, a key advantage of local fuzzy semantics is that it is sensitive to relative logit margins without saturating under changes in the multiplicative logit scale, which we formalize as follows:

**Proposition 3.2 (Scale-invariant dependence on relative logit margins).** *Let $H_t^{se}$ denote the predictive Shannon entropy of the softmax distribution and $H_t^f$ the local fuzzy semantics defined above (with $\epsilon = 0$ for simplicity). For any logits $\mathbf{z}_t = (z_{t,v})_{v \in \mathcal{S}_t}$ with $z_{t,\max} > z_{t,\min}$ and any $a > 0$, $b \in \mathbb{R}$, we have:*

$$H_t^f(a\mathbf{z}_t + b\mathbf{1}) = H_t^f(\mathbf{z}_t),$$

(6)

*whereas $H_t^{se}$ is monotonically decreasing in $a$ and satisfies $\lim_{a \to \infty} H_t^{se} = 0$.*

*Proof.* For local fuzzy semantics, note that for any $a > 0$ and $b \in \mathbb{R}$,

$$\tilde{z}_{t,v} = az_{t,v} + b,$$
$$\tilde{z}_{t,\max} = az_{t,\max} + b, \quad \tilde{z}_{t,\min} = az_{t,\min} + b,$$

(7)

so the normalized memberships satisfy:

$$\tilde{\mu}_{t,v} = \frac{\tilde{z}_{t,v} - \tilde{z}_{t,\min}}{\tilde{z}_{t,\max} - \tilde{z}_{t,\min}} = \frac{az_{t,v} + b - (az_{t,\min} + b)}{az_{t,\max} + b - (az_{t,\min} + b)}$$

$$= \frac{z_{t,v} - z_{t,\min}}{z_{t,\max} - z_{t,\min}} = \mu_{t,v}. \quad (8)$$

Thus the membership profile $\boldsymbol{\mu}_t$ is invariant under any positive affine transformation of the logits, and so is $H_t^f$.

However, for predictive Shannon entropy, let $Z(a) = \sum_{v \in \mathcal{S}_t} \exp(az_{t,v})$ be the partition function. The entropy can be written as:

$$H_t^{se}(a) = \log Z(a) - a \sum_{v \in \mathcal{S}_t} p_v(a) z_{t,v} \quad (9)$$

$$= \log Z(a) - a \mathbb{E}_{p(a)}[z_t].$$

Using the property that $\frac{\mathrm{d}}{\mathrm{d}a} \log Z(a) = \mathbb{E}_{p(a)}[z_t]$, the derivative with respect to $a$ is:

$$\frac{\mathrm{d}}{\mathrm{d}a} H_t^{se}(a) = \frac{\mathrm{d}}{\mathrm{d}a} \log Z(a) - \left( \mathbb{E}_{p(a)}[z_t] + a \frac{\mathrm{d}}{\mathrm{d}a} \mathbb{E}_{p(a)}[z_t] \right)$$

$$= -a \frac{\mathrm{d}}{\mathrm{d}a} \mathbb{E}_{p(a)}[z_t] = -a \operatorname{Var}_{p(a)}(z_t). \quad (10)$$

Since the variance $\operatorname{Var}_{p(a)}(z_t)$ is strictly positive for non-degenerate logits and $a > 0$, we have $\frac{\mathrm{d}}{\mathrm{d}a} H_t^{se}(a) < 0$, implying monotonicity. For the limit, as $a \to \infty$, the probability mass concentrates on the token(s) with the maximum logit value, *i.e.*, $p_v(a) \to 1$ if $v = \arg\max z_{t,v}$ and 0 otherwise. Thus, the distribution converges to a Kronecker delta (one-hot vector), and $H_t^{se}(a) \to 0$. $\square$

Therefore, Shannon entropy saturates as the multiplicative logit scale grows and becomes insensitive to further sharpening of the distribution, whereas local fuzzy semantics remains invariant and depends only on the relative margins between candidates. This scale invariance allows $H_t^f$ to preserve meaningful differences in local ambiguity under arbitrary positive affine transformations of the logits, providing a more robust signal for token-level uncertainty than predictive Shannon entropy. In addition, we present detailed discussions with other evaluation baselines in Appendix A.

### 3.3. POS-aware Token Selection

After obtaining the local fuzzy semantics of each token, a common and straightforward aggregation strategy is averaging token-level entropies across all positions. However, empirically, we observe that tokens with specific part-of-speech (POS) tags exhibit substantially larger fuzzy semantics and align well with diverse hallucination types (objects, attributes, relations, *etc.*). This suggests that sentence-level hallucination risk should be driven primarily by a small subset of linguistically salient tokens.

**POS-aware content tokens.** To locate the aforementioned informative tokens, we adopt a simpler yet effective strategy based on POS tags. For each candidate response $y^{(i)}$, we employ a lightweight POS tagger (Honnibal & Johnson, 2015) and obtain a tag for every token. We then define the index set of *content tokens* as:

$$\mathcal{T}^{(i)} = \big\{ t \in \{1, \dots, T_i\} \,\big|\, pos(y_t^{(i)}) \in$$
$$\{\texttt{NOUN}, \texttt{PROPN}, \texttt{NUM}, \texttt{ADJ}, \texttt{VERB}, \texttt{ADV}, \texttt{ADP}\}\big\}, \quad (11)$$

corresponding to nouns, proper nouns, numbers, adjectives, verbs, adverbs, and adpositions, which carry most factual and relational content in natural language. For each $t \in \mathcal{T}^{(i)}$, we reuse the local fuzzy semantics $H_t^f$ defined in Section 3.2, and this yields a set of POS-filtered fuzzy semantics $\big\{ H_t^f \,\big|\, t \in \mathcal{T}^{(i)} \big\}$, which focuses on tokens where hallucinations are most likely to occur.

**Top-$R$ token pooling.** Even among content tokens, hallucinations tend to concentrate on a few particularly unstable positions (Duan et al., 2024). To further concentrate on these tokens, we sort the POS-filtered fuzzy semantics in descending order:

$$H_{(1)}^f \geq H_{(2)}^f \geq \cdots \geq H_{(|\mathcal{T}^{(i)}|)}^f, \quad (12)$$

and aggregate only the largest $R$ values. The resulting *POS-aware local fuzzy semantics* for $y^{(i)}$ is:

$$U_{pos}(y^{(i)}) = \frac{1}{\min(R, |\mathcal{T}^{(i)}|)} \sum_{r=1}^{\min(R, |\mathcal{T}^{(i)}|)} H_{(r)}^f. \quad (13)$$

When $|\mathcal{T}^{(i)}| < R$, we simply average over all available content tokens. Conceptually, $U_{pos}$ can be seen as a lightweight surrogate for importance-weighted aggregation. This design preserves the self-contained nature of RLSF-V, incurs negligible overhead on top of base generation, and empirically yields a much tighter correlation with hallucination severity than uniform averaging.

### 3.4. Fuzzy Semantic Self-Feedback

Building on the POS-aware local fuzzy semantics $U_{pos}$ defined in Section 3.3, RLSF-V turns these scores into preference signals. Specifically, for each multimodal prompt consisting of an image–text pair $(m, x)$, we obtain a set of candidate responses $\{y^{(i)}\}_{i=1}^N$ together with their fuzzy semantics $\{U_{pos}(y^{(i)})\}_{i=1}^N$ given by (13). We interpret smaller fuzzy semantics as a proxy for higher factual reliability and construct preference pairs by contrasting the most and least certain responses. Concretely, we define:

$$y^{cho} = \arg\min_i \{U_{pos}(y^{(i)})\}_{i=1}^N,$$
$$y^{rej} = \arg\max_i \{U_{pos}(y^{(i)})\}_{i=1}^N, \quad (14)$$

and form the preference pair $\langle y^{cho} \succ y^{rej} \rangle$, indicating that $y^{cho}$ is preferred over $y^{rej}$ for prompt $(m, x)$. Aggregating over prompts yields a preference dataset:

$$\mathcal{D}_f = \left\{ (m, x, y^{cho}, y^{rej}) \right\}. \tag{15}$$

After that, let $\pi_\theta$ denote the trainable policy and $\pi_{\text{ref}}$ a fixed reference policy, for each $(m, x, y^{cho}, y^{rej})$, we optimize a DPO objective that increases the relative likelihood of $y^{cho}$ over $y^{rej}$ under $\pi_\theta$ compared with $\pi_{\text{ref}}$, leading to the fuzzy semantic self-feedback loss:

$$\mathcal{L}_{\text{DPO}}(\theta) = -\mathbb{E}_{(m,x,y^{cho},y^{rej}) \sim \mathcal{D}_f} \Big[ \log \sigma \Big( \beta \log \frac{\pi_\theta(y^{cho} \mid m, x)}{\pi_{\text{ref}}(y^{cho} \mid m, x)} \\ - \beta \log \frac{\pi_\theta(y^{rej} \mid m, x)}{\pi_{\text{ref}}(y^{rej} \mid m, x)} \Big) \Big], \tag{16}$$

where $\sigma(\cdot)$ is the sigmoid function and $\beta > 0$ is a temperature parameter controlling the sharpness of the preference.

By minimizing $\mathcal{L}_{\text{DPO}}$, the updated policy $\pi_\theta$ is encouraged to assign higher likelihood to responses with low $U_{pos}$ and lower likelihood to responses with high $U_{pos}$. In this way, RLSF-V steers the model away from hallucination-prone behaviors without ever querying external models or human annotators.

## 4. Experiments

### 4.1. Experimental Setup

**Implementation Details.** Following prior work (Liu et al., 2025; Peng et al., 2025; Wu et al., 2025; Sarkar et al., 2025; Yu et al., 2025b), we primarily adopt the 7B and 13B variants of LLaVA-v1.5 (Liu et al., 2023) as base models for comparison with existing hallucination mitigation baselines. To further examine the generality of RLSF-V across different MLLM architectures, we additionally apply it to three recent open-source backbones, including Qwen2.5-VL-3B-Instruct (Bai et al., 2025b), Qwen3-VL-8B-Instruct (Bai et al., 2025a), and InternVL3.5-8B-HF (Wang et al., 2025b). To implement RLSF-V, following (Wu et al., 2025), we first randomly sample 10k image–question pairs from RLAIF-V (Yu et al., 2025b) as our training corpus, which contains diverse image-description instructions collected from multiple datasets. For each prompt, we generate $N = 10$ candidate responses with a temperature of $1.0$, top-$p$ of $0.9$, and top-$k$ of $50$ to ensure sufficient diversity. We then apply our fuzzy semantic self-feedback framework with a backbone-specific top-$R$ pooling hyperparameter, setting $R$ to $30$, $20$, $5$, and $40$ for LLaVA-v1.5, Qwen2.5-VL, Qwen3-VL, and InternVL3.5, respectively. The resulting pairwise preference data are used to fine-tune the model with LoRA (Hu et al., 2022). Additional experimental details are provided in Appendix B.1.

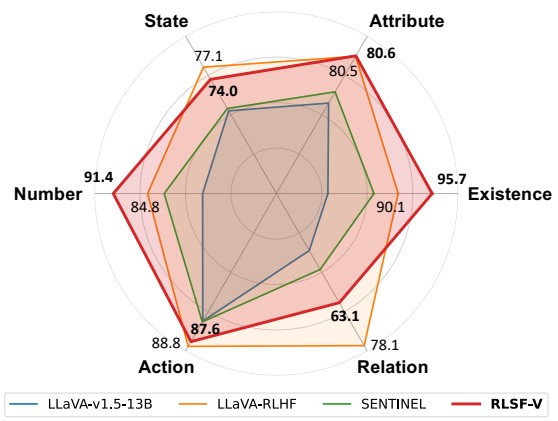

*Figure 3.* AMBER Discriminative per-type F1 scores for the base model (LLaVA-v1.5-13B), an RLHF-style variant (LLaVA-RLHF), an RLAIF-style variant (SENTINEL), and our self-feedback method RLSF-V, which achieves competitive performance across most hallucination types without relying on external feedback.

**Benchmarks.** Our evaluation covers both hallucination-specific benchmarks and general multimodal benchmarks. For hallucination mitigation, we use AMBER Generative (Wang et al., 2023) and Object HalBench (Rohrbach et al., 2018) to evaluate generative hallucinations, and HallusionBench (Guan et al., 2024) and AMBER Discriminative (Wang et al., 2023) to evaluate discriminative hallucinations. To assess whether RLSF-V preserves general multimodal capabilities, we further evaluate on a diverse set of general benchmarks, including MME (Fu et al., 2026), MMMU (Yue et al., 2024), MMStar (Chen et al., 2024c), AI2D (Kembhavi et al., 2016), RealWorldQA (xAI, 2024), ChartQA (Masry et al., 2022), OCRBench (Liu et al., 2024e), SeedBench (Li et al., 2023), and DocVQA (Mathew et al., 2021).

**Baselines.** For the LLaVA-v1.5 backbones, to enable a fair and comprehensive assessment of RLSF-V, we compare it against strong state-of-the-art baselines built on the same backbones. Specifically, we consider 11 methods fine-tuned from LLaVA-v1.5-7B/13B (Liu et al., 2023), including HA-DPO (Zhao et al., 2023), POVID (Zhou et al., 2024), LLaVA-RLHF (Sun et al., 2024), TPO (He et al., 2024b), AMP-IG (Zhang et al., 2024), AMP-MEG (Zhang et al., 2024), HALVA (Sarkar et al., 2025), RLAIF-V (Yu et al., 2025b), OPA-DPO (Wu et al., 2025), SENTINEL (Peng et al., 2025), and symMPO (Liu et al., 2025). Among these, only LLaVA-RLHF constructs its preference dataset from human feedback, whereas all other methods rely on AI feedback from external models, including proprietary systems such as GPT-4V (Achiam et al., 2023), open-source MLLMs such as LLaVA-Next (Liu et al., 2024b) and DeepSeek-V3 (Liu et al., 2024a), and vision–language models such as CLIP (Radford et al., 2021) and Grounding DINO (Liu

*Table 1.* Comparison of hallucination mitigation performance on four benchmarks. For each metric, the best result is shown in **bold** and the second best is underlined. All methods are evaluated under our unified evaluation pipeline using their officially released checkpoints to ensure a fair comparison.

| Model | Size | Feedback | AMBER GEN | | | HallusionBench | Object HalBench | | AMBER DIS | |
|---|---|---|---|---|---|---|---|---|---|---|
| | | | CHAIR↓ | HalRate.↓ | Cog.↓ | All Acc.↑ | Resp.↓ | Ment.↓ | Acc.↑ | F1.↑ |
| LLaVA-v1.5-7B (Liu et al., 2023) | ✗ | ✗ | 7.7 | 35.9 | 4.3 | 42.7 | 25.2 | 46.7 | 71.5 | 74.1 |
| +HA-DPO (Zhao et al., 2023) | 6k | GPT-4 | 6.7 | 30.5 | 3.2 | 43.7 | 20.3 | 36.7 | 74.2 | 78.0 |
| +POVID (Zhou et al., 2024) | 17k | GPT-4V | 9.8 | 51.4 | 7.0 | 42.2 | 25.4 | 52.7 | 72.4 | 74.8 |
| +LLaVA-RLHF (Sun et al., 2024) | 122k | Human | 8.3 | 41.4 | 4.8 | 38.9 | 28.0 | 47.0 | 77.9 | 83.2 |
| +TPO (He et al., 2024b) | 21.4k | LLaVA-Next | 3.6 | 20.5 | 1.6 | 31.6 | 4.4 | 8.0 | 78.8 | 86.2 |
| +AMP-IG (Zhang et al., 2024) | 11k | CLIP | 17.6 | 73.8 | 9.1 | 38.2 | 26.7 | 43.7 | 74.2 | 80.8 |
| +AMP-MEG (Zhang et al., 2024) | 90k | LLaVA & CLIP | 13.2 | 59.1 | 5.2 | 37.0 | 25.7 | 32.3 | 73.7 | 77.8 |
| +HALVA (Sarkar et al., 2025) | 21.5k | GPT-4V | 7.0 | 33.6 | 3.6 | 44.2 | 23.8 | 43.0 | 77.9 | 83.4 |
| +RLAIF-V (Yu et al., 2025b) | 83k | LLaVA-Next | 2.9 | 15.6 | 0.9 | 28.7 | 5.6 | 9.7 | 54.3 | 73.9 |
| +OPA-DPO (Wu et al., 2025) | 4.8k | GPT-4V | 2.5 | 12.1 | 0.9 | 42.2 | 3.8 | 5.7 | 80.5 | 85.1 |
| +SENTINEL (Peng et al., 2025) | 8.7k | G-DINO & Yolo-W | 3.0 | 14.8 | 1.3 | 42.9 | 3.0 | 5.0 | 76.1 | 79.3 |
| +symMPO (Liu et al., 2025) | 21.4k | DeepSeek-V3 | 5.9 | 28.3 | 3.6 | 45.3 | 9.6 | 15.7 | **81.2** | **86.5** |
| **+RLSF-V** | 10k | self-feedback | **2.4**$_{\downarrow 5.3}$ | **5.1**$_{\downarrow 30.8}$ | **0.3**$_{\downarrow 4.0}$ | **47.4**$_{\uparrow 4.7}$ | **2.9**$_{\downarrow 22.3}$ | **4.0**$_{\downarrow 42.7}$ | 78.8$_{\uparrow 7.3}$ | 84.0$_{\uparrow 9.9}$ |
| LLaVA-v1.5-13B (Liu et al., 2023) | ✗ | ✗ | 6.9 | 32.0 | 3.5 | 43.3 | 22.6 | 45.0 | 71.3 | 73.1 |
| +LLaVA-RLHF (Sun et al., 2024) | 122k | Human | 6.9 | 34.4 | 3.7 | 42.3 | 20.9 | 41.0 | 81.7 | 85.2 |
| +AMP-IG (Zhang et al., 2024) | 90k | CLIP | 13.4 | 52.0 | 5.4 | 40.2 | 33.8 | 57.7 | 66.2 | 70.3 |
| +AMP-MEG (Zhang et al., 2024) | 11k | LLaVA & CLIP | 11.6 | 51.1 | 5.4 | 39.9 | 24.9 | 38.7 | 62.8 | 63.4 |
| +HALVA (Sarkar et al., 2025) | 21.5k | GPT-4V | 6.5 | 30.6 | 3.1 | 46.2 | 19.2 | 39.0 | 82.9 | 86.5 |
| +OPA-DPO (Wu et al., 2025) | 4.8k | GPT-4V | 2.7 | 13.4 | 0.9 | 42.2 | 4.9 | 7.3 | 84.1 | 87.5 |
| +SENTINEL (Peng et al., 2025) | 7k | G-DINO & Yolo-W | 2.7 | 12.0 | 0.9 | 43.8 | 2.4 | 4.0 | 76.2 | 78.7 |
| +symMPO (Liu et al., 2025) | 21.4k | DeepSeek-V3 | 4.6 | 26.7 | 2.4 | 45.1 | 8.4 | 16.0 | 84.5 | **89.0** |
| **+RLSF-V** | 10k | self-feedback | **2.0**$_{\downarrow 4.9}$ | **4.4**$_{\downarrow 27.6}$ | **0.3**$_{\downarrow 3.2}$ | **48.1**$_{\uparrow 4.8}$ | **1.2**$_{\downarrow 21.4}$ | **1.7**$_{\downarrow 43.3}$ | **84.8**$_{\uparrow 13.5}$ | 87.6$_{\uparrow 14.5}$ |
| Qwen2.5-VL-3B-Instruct (Bai et al., 2025b) | ✗ | ✗ | 8.0 | 49.1 | 5.2 | 53.1 | 9.7 | 15.7 | 88.0 | 91.0 |
| **+RLSF-V** | 10k | self-feedback | **6.5**$_{\downarrow 1.5}$ | **36.5**$_{\downarrow 12.6}$ | **3.9**$_{\downarrow 1.3}$ | **53.9**$_{\uparrow 0.8}$ | **7.0**$_{\downarrow 2.7}$ | **9.3**$_{\downarrow 6.4}$ | **88.2**$_{\uparrow 0.2}$ | **91.1**$_{\uparrow 0.1}$ |
| Qwen3-VL-8B-Instruct (Bai et al., 2025a) | ✗ | ✗ | 8.1 | 62.0 | 3.4 | 64.0 | 8.1 | 14.3 | 89.0 | 91.7 |
| **+RLSF-V** | 10k | self-feedback | **6.7**$_{\downarrow 1.4}$ | **48.1**$_{\downarrow 13.9}$ | **2.3**$_{\downarrow 1.1}$ | **65.5**$_{\uparrow 1.5}$ | **5.5**$_{\downarrow 2.6}$ | **9.7**$_{\downarrow 4.6}$ | **89.5**$_{\uparrow 0.5}$ | **92.1**$_{\uparrow 0.4}$ |
| InternVL3.5-8B-HF (Wang et al., 2025b) | ✗ | ✗ | 7.6 | 63.8 | 5.3 | 57.6 | 5.5 | 9.4 | 86.7 | 89.4 |
| **+RLSF-V** | 10k | self-feedback | **7.2**$_{\downarrow 0.4}$ | **58.7**$_{\downarrow 5.1}$ | **4.5**$_{\downarrow 0.8}$ | **58.1**$_{\uparrow 0.5}$ | **4.4**$_{\downarrow 1.1}$ | **6.7**$_{\downarrow 2.7}$ | **86.7**$_{\uparrow 0.0}$ | **89.4**$_{\uparrow 0.0}$ |

*Table 2.* General multimodal benchmark performance before and after applying RLSF-V to Qwen3-VL-8B-Instruct.

| Benchmark | Qwen3-VL-8B-Instruct | +RLSF-V |
|---|---|---|
| MME-Cog. (Fu et al., 2026) | **641.79** | 638.93$_{\downarrow 2.86}$ |
| MME-Per. (Fu et al., 2026) | **1720.80** | 1715.30$_{\downarrow 5.50}$ |
| MMMU (Yue et al., 2024) | 52.33 | **52.89**$_{\uparrow 0.56}$ |
| MMStar (Chen et al., 2024c) | 62.88 | **63.78**$_{\uparrow 0.90}$ |
| AI2D (Kembhavi et al., 2016) | 83.65 | **83.71**$_{\uparrow 0.06}$ |
| RealWorldQA (xAI, 2024) | 69.41 | **70.20**$_{\uparrow 0.79}$ |
| ChartQA (Masry et al., 2022) | 85.36 | **85.52**$_{\uparrow 0.16}$ |
| OCRBench (Liu et al., 2024e) | **83.00** | 82.40$_{\downarrow 0.60}$ |
| SeedBench (Li et al., 2023) | 75.07 | **75.15**$_{\uparrow 0.08}$ |
| DocVQA (Mathew et al., 2021) | 95.61 | **95.95**$_{\uparrow 0.34}$ |

et al., 2024d). In addition to these LLaVA-based baselines, we also compare each newly adapted backbone with its corresponding base model to evaluate the architecture-level transferability of RLSF-V.

### 4.2. Comparison with State-of-the-arts

To ensure a fair comparison, we re-evaluate all baselines by running inference with their officially released checkpoints on all benchmarks, and adopt deterministic decoding with the temperature set to 0 for every method, including ours. The hallucination mitigation results are summarized in Table 1 and Figure 3, while the general multimodal benchmark results are reported in Table 2. From these results, we make

the following observations.

**(1) Strong hallucination suppression.** RLSF-V achieves the best or second-best performance on most benchmarks, and substantially reduces hallucination-related metrics compared to the strongest baselines. For example, on AMBER Generative, LLaVA-v1.5-7B+RLSF-V reduces the hallucination rate (HalRate.) by about 7% compared to OPA-DPO (Wu et al., 2025), and on HallusionBench, it improves the overall question-answering accuracy by 2.1% over symMPO (Liu et al., 2025).

**(2) Moderate data scale with strong scalability.** Although RLSF-V does not attain the best performance using the smallest amount of training data, it does not rely on any external feedback signals. In contrast to all other baselines, which require either costly human annotations, APIs, or calls to external large models, RLSF-V only uses self-generated fuzzy semantic feedback. This makes RLSF-V naturally scalable to arbitrary multimodal backbones without redesigning the training pipeline (*e.g.*, Qwen2.5-VL-3B-Instruct, Qwen3-VL-8B-Instruct, and InternVL3.5-8B-HF).

**(3) Analysis on AMBER Discriminative.** Figure 3 further provides a fine-grained comparison on AMBER Discriminative among the base model (LLaVA-v1.5), the human-feedback-based LLaVA-RLHF (Sun et al., 2024), the external model feedback based SENTINEL (Peng et al., 2025),

*Table 3.* Component-wise ablation of RLSF-V with LLaVA-v1.5-7B and 13B backbones.

| Model | Type | AMBER GEN | | | HallusionBench | Object HalBench | | AMBER DIS | |
|---|---|---|---|---|---|---|---|---|---|
| | | CHAIR↓ | HalRate.↓ | Cog.↓ | All Acc.↑ | Resp.↓ | Ment.↓ | Acc.↑ | F1.↑ |
| LLaVA-v1.5-7B | RLSF-V | **2.4** | **5.1** | **0.3** | **47.4** | **2.9** | **4.0** | 78.8 | **84.0** |
| | W/O POS-aware Token Selection | 2.4 | 5.9 | 0.4 | 44.5 | 3.1 | 4.7 | 76.3 | 79.2 |
| | W/O Top-R token pooling | 16.5 | 50.3 | 3.6 | 43.5 | 41.9 | 70.3 | 78.4 | 83.7 |
| LLaVA-v1.5-13B | RLSF-V | **2.0** | **4.4** | **0.3** | **48.1** | **1.2** | **1.7** | **84.8** | **87.6** |
| | W/O POS-aware Token Selection | 2.5 | 6.1 | 0.4 | 44.3 | 2.2 | 3.3 | 63.3 | 62.4 |
| | W/O top-R token pooling | 15.9 | 56.2 | 4.2 | 45.3 | 41.2 | 69.3 | 82.2 | 86.0 |

*Table 4.* Comparison of different self-feedback within RLSF-V.

| Model | Feedback | AMBER GEN | | | Object HalBench | |
|---|---|---|---|---|---|---|
| | | CHAIR↓ | HalRate.↓ | Cog.↓ | Resp.↓ | Ment.↓ |
| LLaVA-v1.5-7B | ✗ | 7.7 | 35.9 | 4.3 | 25.2 | 46.7 |
| | Energy | 14.5 | 46.7 | 6.1 | 22.0 | 46.7 |
| | Evidence | 4.2 | 10.8 | 0.9 | 7.6 | 13.7 |
| | Entropy | 2.8 | 5.8 | **0.3** | **2.9** | **4.0** |
| | Fuzzy | **2.4** | **5.1** | **0.3** | **2.9** | **4.0** |
| LLaVA-v1.5-13B | ✗ | 6.9 | 32.0 | 3.5 | 22.6 | 45.0 |
| | Energy | 6.4 | 16.9 | 2.3 | 8.5 | 14.7 |
| | Evidence | **1.7** | **3.2** | **0.3** | 3.7 | 5.3 |
| | Entropy | 2.3 | 5.8 | **0.3** | 2.5 | 3.7 |
| | Fuzzy | 2.0 | 4.4 | **0.3** | **1.2** | **1.7** |

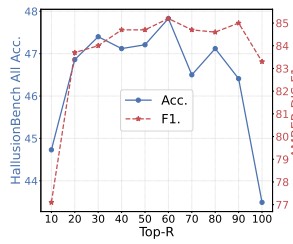 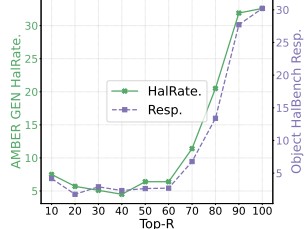

*(a)* Discrimative benchmark  *(b)* Generative benchmark

*Figure 4.* Effect of the top-$R$ pooling strategy on four benchmark performances using LLaVA-v1.5-7B as the backbone. Both plots show that a moderate value (around $R = 30$) leads to competitive hallucination mitigation, whereas very small or very large $R$ degrades performance.

and our RLSF-V. For the hallucination types *Existence*, *Attribute*, *State*, *Number*, and *Action*, the performance of RLSF-V is comparable to or even surpasses that of LLaVA-RLHF. We attribute this to our POS-aware token selection strategy, which explicitly focuses on content words whose parts of speech are closely aligned with these hallucination types, combined with our fuzzy semantic self-feedback mechanism that suppresses the corresponding hallucinations. However, for the *Relation* type, RLSF-V clearly lags behind LLaVA-RLHF. We hypothesize that relation hallucinations are harder to reliably assess based on prepositions and related tokens (even though they are included in our token selection), because such tokens are often used in non-relational contexts, leading to weaker or noisier signals for relation-specific hallucinations.

**(4) Preservation of general multimodal capabilities.** Besides hallucination-specific benchmarks, we further examine whether RLSF-V affects the general capabilities of the backbone model. As shown in Table 2, applying RLSF-V to Qwen3-VL-8B-Instruct preserves competitive performance across a broad range of general multimodal benchmarks. RLSF-V improves results on 7 out of 10 benchmarks, including MMMU, MMStar, AI2D, RealWorldQA, ChartQA, SeedBench, and DocVQA, while only causing small drops on MME-Cognition, MME-Perception, and OCRBench. These results indicate that RLSF-V mitigates hallucinations without substantially sacrificing the general multimodal understanding and reasoning ability of the base model.

### 4.3. Ablation Studies

**Component effectiveness.** Table 3 presents the performance of RLSF-V with and without the POS-aware token selection and the top-$R$ token pooling, evaluated on both the LLaVA-v1.5-7B and LLaVA-v1.5-13B backbones. Across four representative benchmarks, removing the POS-aware token selection module leads to a slight degradation in performance. We attribute this to the fact that, without POS-aware filtering, the fuzzy semantics estimation at the token level involves many irrelevant tokens, which dilutes the contribution of the truly informative tokens that are critical for hallucination assessment. In contrast, removing the top-$R$ token pooling strategy causes a much more pronounced drop in performance. In this variant, all extracted tokens are aggregated when computing the sentence-level fuzzy semantic score, which strongly dilutes hallucination-related signals. This observation is consistent with prior findings that hallucinations are typically localized to a small subset of tokens within a sentence rather than being uniformly distributed across the sequence (Chen et al., 2024d).

**Effectiveness of hallucination evaluation.** The accuracy of hallucination assessment directly determines the quality of the constructed preference dataset, which in turn governs the effectiveness of the subsequent direct preference optimization. Therefore, a key question for self-feedback methods is whether the underlying hallucination evaluation mecha-

nism is effective. In Table 4, we compare our local fuzzy semantics self-feedback with several classical alternatives that can be used for hallucination assessment, including energy-based (Liu et al., 2020), evidence theory-based (Ma et al., 2025a), and Shannon-entropy-based feedback (Kadavath et al., 2022). For a fair comparison, we keep the POS-aware token selection and the top-$R$ token pooling strategy unchanged, and only replace the evaluation module within RLSF-V. The results show that our local fuzzy semantics of internal model signals achieves the best downstream fine-tuning performance in most cases. This indicates that local ambiguity between competing continuations provide a more discriminative and robust estimate of hallucination-related uncertainty, yielding higher-quality self-generated preference data and thereby leading to more effective hallucination mitigation.

### 4.4. Parameter Analysis

We further examine the sensitivity of RLSF-V to the top-$R$ pooling hyperparameter in Equation (13). Figure 4 reports the results on LLaVA-v1.5-7B when varying $R$ from 10 to 100 on both discriminative and generative benchmarks. On the discriminative side (Figure 4a), performance improves rapidly when increasing $R$ from 10 to around 30–40, but degrades again as $R$ grows larger. A similar pattern is observed on the generative benchmarks (Figure 4b), where smaller hallucination rates are obtained for moderate $R$, while very large $R$ leads to sharply increased hallucination rates. These trends suggest that using too few tokens makes the uncertainty estimate overly sensitive to a small number of noisy positions, whereas averaging over too many tokens dilutes hallucination-related signals.

## 5. Conclusion

In this paper, we presented RLSF-V, a self-feedback framework that leverages fuzzy semantics derived from a model's internal logits to mitigate hallucinations in MLLMs. Unlike traditional RLHF- or RLAIF-based pipelines, RLSF-V does not rely on any external large models or human annotations for feedback. Experiments on multiple benchmarks demonstrate that RLSF-V achieves competitive or superior hallucination mitigation compared to RLHF/RLAIF baselines, indicating that internally generated fuzzy semantic signals can serve as an effective alternative to external evaluators and can be readily applied across different multimodal backbones. In future work, we plan to extend this line of self-feedback methods to purely textual LLMs and other feedback-driven learning setups, further exploring how internal uncertainty signals can be exploited to improve robustness and alignment while maintaining strong task performance.

## Acknowledgements

This work was supported in part by the National Natural Science Foundation of China under Grant U25B6003, 62472295, and U25A20534; in part by Foundation Enhancement Program Project (Technology Field Fund) under Grant 2025-JCJQ-JJ-0686; in part by Sichuan Science and Technology Planning Project under Grant 24NSFTD0130; in part by the Luzhou City School-Local-Enterprise-Academy Science and Technology Cooperation Project under Grant 2024XDY200; and in part by the Fundamental Research Funds for the Central Universities under Grant CJ202303 and CJ202403.

## Impact Statement

This paper presents work whose goal is to advance the field of machine learning. There are many potential societal consequences of our work, none of which we feel must be specifically highlighted here.

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

# Appendix

In this appendix, we provide additional material that complements the main paper, *RLSF-V: Mitigating Hallucinations in MLLMs via Fuzzy Semantic Self-Feedback*. Specifically, we include the following contents:

- In Appendix B, we provide additional experimental details, including:
    - Training setup (in Appendix B.1);
    - Descriptions of the datasets and evaluation metrics (in Appendix B.2);
    - Pseudocode for RLSF-V (in Appendix B.3).

- In Appendix D, we provide qualitative case studies of RLSF-V, including:
    - Examples of constructing preference pairs using self-feedback (in Appendix D.1);
    - Examples of hallucination mitigation during inference (in Appendix D.2).

## A. Self-Feedback Baseline Details

In this section, we detail the uncertainty-aware self-feedback baselines reported in Table 4 of the main paper. All methods operate at the token level, *i.e.*, given the model's logits at decoding step $t$, they compute a scalar uncertainty score $U_t$ that is then aggregated with POS-aware Token Selection as described in the main paper.

### A.1. Energy-based self-feedback

We consider an energy-based construction that measures uncertainty directly from logits via the log-partition function (Liu et al., 2020):

$$U_t^{eng} = \log \sum_{v \in \mathcal{V}} \exp(z_{t,v}). \tag{19}$$

This quantity is closely related to the log-normalizer of the corresponding Gibbs distribution and grows as the predictive distribution becomes flatter. When probability mass is distributed over many tokens, the sum $\sum_v \exp(z_{t,v})$ increases and so does $U_t^{eng}$. Conversely, when one logit dominates, the sum is dominated by a single term and $U_t^{eng}$ becomes smaller.

### A.2. Evidence-based self-feedback

Finally, we adopt an evidential formulation that explicitly decomposes predictive uncertainty into epistemic and aleatoric components (Sensoy et al., 2018). We first convert logits into nonnegative evidence. Next, we restrict attention to the Top-$K$ candidate tokens at each step (by logit value) and denote the restricted logits by $\mathbf{z}_t^{(K)} \in \mathbb{R}^K$; when $K$ is unspecified, we set $K = 50$. Evidence is defined as:

$$\mathbf{e}_t = \mathrm{ReLU}\big(\mathbf{z}_t^{(K)}\big), \qquad S_t = \sum_{j=1}^{K} e_{t,j}, \tag{20}$$

where $e_{t,j} \geq 0$ is the evidence associated with the $j$-th candidate and $S_t$ is the total evidence at step $t$.

Based on $S_t$, we estimate an epistemic uncertainty term:

$$\mathrm{EU}_t = \frac{K}{S_t + K}, \tag{21}$$

which is large when the total evidence is small (indicating limited support from the model) and decreases as $S_t$ grows. To capture aleatoric uncertainty, we use a digamma-based measure that reflects how the evidence is distributed across candidates:

$$\text{AU}_t = \sum_{j=1}^{K} \left( -\frac{e_{t,j}}{S_t} \left[ \psi(e_{t,j} + 1) - \psi(S_t + 1) \right] \right), \tag{22}$$

where $\psi(\cdot)$ denotes the digamma function. Intuitively, $\text{AU}_t$ is higher when the evidence vector $\mathbf{e}_t$ is more dispersed across multiple tokens, indicating greater inherent ambiguity among plausible continuations. The final evidence-based uncertainty score combines these two components multiplicatively (Ma et al., 2025a):

$$U_t^{evi} = \text{EU}_t \cdot \text{AU}_t. \tag{23}$$

Therefore, this baseline assigns high uncertainty when the model supplies little overall evidence (high $\text{EU}_t$) and when the available evidence is spread across many competing tokens (high $\text{AU}_t$).

### A.3. Discussion

Compared to these uncertainty-aware baselines, our local fuzzy semantics offers a more direct and robust characterization of token-level ambiguity. To be specific, the energy-based score $U_t^{eng}$ and the evidential score $U_t^{evi}$ both operate in logit space but remain strongly dependent on the multiplicative logit scale, the vocabulary size, and additional design choices. In contrast, our fuzzy semantics is defined on min–max normalized logits and is invariant under any positive affine transformation, so it depends solely on the *relative* margins between candidates. This scale invariance allows local fuzzy semantics to remain stable across different models and decoding temperatures, while being particularly sensitive to situations where the model hesitates among several semantically plausible tokens—precisely the regime in which hallucinations and factual errors tend to arise.

## B. Experimental Details

### B.1. Training Setup

Our main experiments are conducted on the 7B and 13B variants of LLaVA-v1.5 (Liu et al., 2023). Both models follow the standard LLaVA-v1.5 architecture, which uses Vicuna-v1.5-7B/13B as the language backbone (Zheng et al., 2023) and CLIP ViT-L/336px as the vision encoder (Radford et al., 2021), connected by a two-layer MLP projector with GELU activation ("mlp2x_gelu"). On top of these backbones, we adopt parameter-efficient fine-tuning with LoRA (Hu et al., 2022) and only train the linear layers in the LLM, while freezing the vision encoder and the projector. For both the 7B and 13B models, we set the LoRA rank to $r = 256$ and the LoRA scaling factor to $\alpha = 512$. For preference learning, we set the temperature parameter in Equation (16) to $\beta = 0.1$. We use AdamW (Loshchilov & Hutter, 2017) as the optimizer with a base learning rate of $4 \times 10^{-6}$, a cosine learning rate schedule, and a weight decay of $0.01$. The maximum text sequence length is set to 2048 tokens, and we apply a warm-up ratio of 0.05 and train for 2 epochs. We further use a batch size of 8 per GPU and a global batch size of 32 with data parallelism. All experiments are run on four NVIDIA RTX 4090 GPUs with 48 GB of memory each, and to efficiently utilize GPU memory and support batch sizes, we employ ZeRO stage 2 optimization (Rajbhandari et al., 2020).

To further evaluate the transferability of RLSF-V across different MLLM architectures, we additionally fine-tune Qwen2.5-VL-3B-Instruct (Bai et al., 2025b), Qwen3-VL-8B-Instruct (Bai et al., 2025a), and InternVL3.5-8B-HF (Wang et al., 2025b). These models are trained using the LLaMA-Factory (Zheng et al., 2024) framework with DPO-based LoRA fine-tuning. For all three backbones, we set the temperature parameter in Equation (16) to $\beta = 0.1$, and set the LoRA rank to $r = 256$ and the LoRA scaling factor to $\alpha = 512$. The maximum sequence length is set to 2048 tokens, and all models are trained for 2 epochs with a cosine learning rate schedule and bfloat16 precision. For Qwen2.5-VL-3B-Instruct, we use a learning rate of $4 \times 10^{-6}$, a warm-up ratio of 0.05, weight decay of 0.01, per-device batch size of 1, and gradient accumulation steps of 16. For Qwen3-VL-8B-Instruct and InternVL3.5-8B-HF, we freeze the vision tower and the multimodal projector, use a learning rate of $5 \times 10^{-6}$, per-device batch size of 1, and gradient accumulation steps of 8. Unless otherwise specified, the same self-feedback data construction pipeline is used for all backbones.

**Implementation of POS-aware token selection.** As described in Section 3.3, our local fuzzy semantics relies on POS-aware content tokens. For completeness, we detail the implementation used in our experiments. Given the subword tokens of

---

**Algorithm 1** RLSF-V: Mitigating Hallucinations in MLLMs via Fuzzy Semantic Self-Feedback

---

**Require:** Base MLLM $\pi_\theta$, multimodal training corpus $\mathcal{D} = \{(m, x)\}$ (image–text prompts), number of candidates $N$, top-$R$ tokens $R$, temperature $\beta$.
**Ensure:** Fine-tuned policy $\pi_\tau$ for hallucination-mitigated generation.

1: *// Diverse candidate generation*
2: **for** each prompt $(m, x) \in \mathcal{D}$ **do**
3:     Sample $N$ candidate responses $\{y^{(i)}\}_{i=1}^N \sim \pi_\theta(\cdot \mid m, x)$ using stochastic decoding.
4:     Cache token-level logits $\{z_t^{(i)}\}$ for later local fuzzy semantics estimation.

5:     *// POS-aware local fuzzy semantics*
6:     **for** each candidate $y^{(i)}$ **do**
7:         For each position $t$, compute the local fuzzy semantics $H_t^{f,(i)}$ from logits $z_t^{(i)}$ using Equation (4) and Equation (5).
8:         Employ the POS tagger on $y^{(i)}$ and obtain the POS-aware content token index set $\mathcal{T}^{(i)}$ as defined in Equation (11).
9:         Collect the POS-filtered content tokens $\{H_t^{f,(i)} \mid t \in \mathcal{T}^{(i)}\}$ and apply the top-$R$ pooling in Equation (13) to obtain the local fuzzy semantics $U_{pos}(y^{(i)})$.
10:     **end for**

11:     *// Fuzzy semantic self-feedback and preference construction*
12:     Determine the chosen and rejected candidates $y^{cho}$ and $y^{rej}$ according to Equation (12), based on $U_{pos}(\cdot)$.
13:     **if** $\left|U_{pos}(y^{cho}) - U_{pos}(y^{rej})\right| > 0$ **then**
14:         Add the ordered preference quadruple $(m, x, y^{cho}, y^{rej})$ to the preference dataset $\mathcal{D}_f$ as in Equation (15).
15:     **end if**
16: **end for**

17: *// Fine-tuning with self-generated preferences*
18: Initialize the policy $\pi_\tau \leftarrow \pi_\theta$ and fix the reference policy $\pi_{\text{ref}} \leftarrow \pi_\theta$.
19: **for** each training step with mini-batch $\mathcal{B} \subseteq \mathcal{D}_f$ **do**
20:     Compute the loss $\mathcal{L}_{\text{DPO}}$ on $\mathcal{B}$ following Equation (16) with temperature $\beta$.
21:     Update $\pi_\tau$ by minimizing $\mathcal{L}_{\text{DPO}}$.
22: **end for**

---

a generated answer, we first reconstruct the answer text and run a POS tagger to obtain word-level part-of-speech tags. We then identify content words whose POS tags fall into the target set in Equation (11) and mark the corresponding character spans in the text. A token is selected as a *content token* if any character in its span overlaps with these marked spans. This yields a binary mask over tokens that filters out POS-irrelevant tokens before computing hallucination-related fuzzy semantics.

### B.2. Datasets and Metrics

**Training Dataset.** To simplify the construction of preference data, we build RLSF-V on top of the instruction dataset provided by RLAIF-V (Yu et al., 2025b). This dataset contains about 83k image–question pairs collected from diverse sources, including MSCOCO (Lin et al., 2014), ShareGPT-4V (Chen et al., 2024b), MovieNet (Huang et al., 2020), Google Landmark v2 (Weyand et al., 2020), VQA v2 (Goyal et al., 2017), OK-VQA (Marino et al., 2019), and TextVQA (Singh et al., 2019). In our work, we do not use the preference pairs released by RLAIF-V (Yu et al., 2025b). Instead, we randomly sample 10k questions from this corpus as prompts and construct our own self-feedback-based preference data on top of them.

**Evaluation Benchmark and Metrics.** To comprehensively evaluate RLSF-V, we employ four specialized benchmarks to measure hallucination mitigation: AMBER Generative, Object HalBench, HallusionBench, and AMBER Discriminative.

- **AMBER Generative** (Wang et al., 2023) is a hallucination benchmark with detailed object annotations on 1,004 images in a generative captioning task. It compares the set of objects mentioned in a model response with the ground-truth objects in the image and defines three metrics. (1) **CHAIR** measures the fraction of hallucinated object mentions among all mentioned objects. (2) **HalRate** is a response-level hallucination rate, *i.e.*, the proportion of responses that

*Table 5.* Comparison of ScienceQA performance. We report accuracy (%) across different categories: Natural Science (NAT), Social Science (SOC), Language Science (LAN), Text Context (TXT), Image Context (IMG), No Context (NO), Grades 1-6 (G1-6), and Grades 7-12 (G7-12). The best result is shown in **bold** and the second best is underlined.

| Model | Size | Feedback | Subject | | | Context Modality | | | Grade | | Avg |
|---|---|---|---|---|---|---|---|---|---|---|---|
| | | | NAT | SOC | LAN | TXT | IMG | NO | G1-6 | G7-12 | |
| LLaVA-v1.5-7B (Liu et al., 2023) | ✗ | ✗ | **70.52** | 74.24 | 67.09 | 70.77 | 69.46 | 67.32 | **73.97** | 64.01 | **70.41** |
| +LLaVA-RLHF (Sun et al., 2024) | 122k | Human | 68.69 | **77.28** | 64.73 | **71.11** | **70.25** | 64.25 | 72.61 | 63.81 | 69.46 |
| +TPO (He et al., 2024b) | 21.4k | LLaVA-Next | 69.85 | 70.42 | 64.82 | 70.04 | 67.13 | 65.09 | 71.44 | 63.68 | 68.66 |
| +AMP-IG (Zhang et al., 2024) | 11k | CLIP | 58.17 | 62.43 | 58.27 | 58.11 | 60.04 | 57.63 | 62.81 | 52.41 | 59.09 |
| +AMP-MEG (Zhang et al., 2024) | 90k | LLaVA & CLIP | 58.97 | 62.65 | 57.00 | 60.22 | 59.54 | 55.12 | 62.96 | 52.54 | 59.23 |
| +RLAIF-V (Yu et al., 2025b) | 83k | LLaVA-Next | 69.49 | 71.54 | 62.00 | 69.60 | 68.12 | 63.14 | 70.89 | 62.76 | 67.98 |
| +OPA-DPO (Wu et al., 2025) | 4.8k | GPT-4V | 70.07 | 73.57 | 67.18 | 70.09 | 68.67 | 67.46 | 73.68 | 63.55 | 70.05 |
| +symMPO (Liu et al., 2025) | 21.4k | DeepSeek-V3 | 70.38 | 72.33 | **67.45** | 70.43 | 68.17 | **67.74** | 73.38 | **64.01** | 70.03 |
| **Ours** | 10k | self-feedback | 68.87 | 72.78 | 66.82 | 67.99 | 67.38 | 67.53 | 72.61 | 62.95 | 69.16 |
| LLaVA-v1.5-13B (Liu et al., 2023) | ✗ | ✗ | **75.22** | 77.05 | **72.64** | 75.61 | 72.83 | 73.45 | **77.72** | 69.94 | **74.94** |
| +LLaVA-RLHF (Sun et al., 2024) | 122k | Human | 73.45 | **78.63** | 70.64 | 74.19 | 72.73 | 72.06 | 76.98 | 68.09 | 73.80 |
| +AMP-IG (Zhang et al., 2024) | 11k | CLIP | 69.09 | 75.03 | 64.73 | 69.79 | 69.16 | 67.04 | 72.32 | 63.61 | 69.21 |
| +AMP-MEG (Zhang et al., 2024) | 90k | LLaVA & CLIP | 66.52 | 73.00 | 63.00 | 68.48 | 68.72 | 62.44 | 70.52 | 60.58 | 66.97 |
| +OPA-DPO (Wu et al., 2025) | 4.8k | GPT-4V | 74.16 | 76.15 | 64.09 | 74.29 | 72.14 | 66.55 | 76.06 | 64.60 | 71.96 |
| +symMPO (Liu et al., 2025) | 21.4k | DeepSeek-V3 | 73.93 | 75.37 | 70.00 | 74.39 | 71.05 | 71.29 | 75.92 | 68.36 | 73.21 |
| **Ours** | 10k | self-feedback | 72.91 | 75.70 | 70.36 | 72.78 | 70.40 | 71.71 | 76.25 | 66.71 | 72.84 |

contain at least one hallucinated object. (3) **Cog** evaluates how often the model produces objects from a predefined list of cognitively plausible hallucinations.

- **Object HalBench** (Rohrbach et al., 2018) focuses on common object hallucinations in detailed image descriptions. Following prior work, we evaluate on the official split containing 300 instances with diverse prompts to obtain stable estimates. We adopt two official metrics: the response-level hallucination rate (**Resp.**), which is the proportion of responses that contain any hallucinated object, and the mention-level hallucination rate (**Ment.**), which measures the percentage of hallucinated object mentions among all object mentions.

- **HallusionBench** (Guan et al., 2024) is a benchmark targeting multimodal hallucination in image–context reasoning. It contains 346 images paired with 1,129 carefully designed prompts that probe both language-only hallucinations and vision-induced illusions. To quantify model performance, we follow the official protocol and report the overall accuracy across all questions (**All Acc**), which aggregates both straightforward and more challenging reasoning cases.

- **AMBER Discriminative** (Wang et al., 2023) is the discriminative component of AMBER, consisting of about 15k fine-grained annotations for evaluating hallucination severity from a classification perspective. Each instance is labeled along six aspects: object existence, attributes, relationships, state, number, and actions. For this benchmark, we evaluate models with two metrics: **Acc**, the overall classification accuracy across all aspects, and **F1**, the F1 score that balances precision and recall of non-hallucinatory predictions.

To examine whether RLSF-V preserves the general multimodal capabilities of the backbone model, we further evaluate on a diverse set of general benchmarks, including MME (Fu et al., 2026), MMMU (Yue et al., 2024), MMStar (Chen et al., 2024c), AI2D (Kembhavi et al., 2016), RealWorldQA (xAI, 2024), ChartQA (Masry et al., 2022), OCRBench (Liu et al., 2024e), SeedBench (Li et al., 2023), DocVQA (Mathew et al., 2021), and ScienceQA (Lu et al., 2022). These benchmarks cover a broad range of capabilities, including perception and cognition evaluation, multi-discipline reasoning, visual question answering, chart and document understanding, OCR-oriented recognition, real-world visual reasoning, and science-domain question answering.

## B.3. Fuzzy Semantic Self-Feedback Pseudocode

We summarize in Algorithm 1 how fuzzy semantic self-feedback is used to construct the preference dataset and how it is integrated with preference optimization to mitigate hallucinations in multimodal large language models.

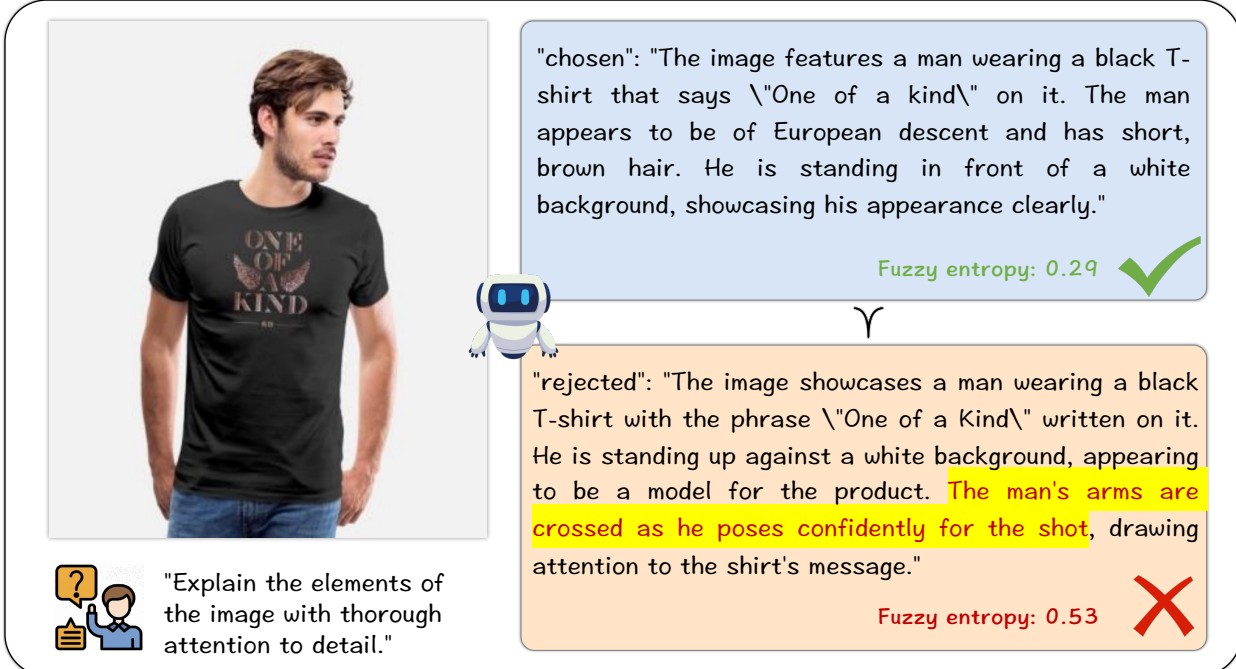

*Figure 5.* Case study of the fuzzy semantics feedback preference pairs.

## C. Experiments on ScienceQA

Table 5 shows that our method achieves *upper-middle* performance on ScienceQA across both model scales of LLaVA. Although it does not obtain the best absolute accuracy, LLaVA-v1.5-7B+RLSF-V is consistently competitive with or close to the strongest systems, reaching $69.16\%$ and $72.84\%$ average accuracy on the 7B and 13B backbones, respectively. Importantly, most higher-scoring baselines rely on costly or less accessible feedback sources, such as human annotations, proprietary alignment models, or large external vision–language models (*e.g.*, CLIP, LLaVA-Next, GPT-4V, DeepSeek-V3) to provide supervisory signals. By contrast, our method derives all training signals from the model's own internal logits, without requiring any additional models or human-in-the-loop labeling, which makes it substantially easier to scale and deploy in practice. Moreover, as shown in our hallucination evaluations in the main paper, these externally supervised baselines offer only limited suppression of hallucinations compared to RLSF-V, whereas our self-feedback scheme yields markedly stronger reductions in hallucination rates while maintaining competitive ScienceQA accuracy.

## D. Case Studies

In this section, we present qualitative case studies to provide an intuitive understanding of how RLSF-V operates in practice and how it mitigates hallucinations in multimodal large language models. We first illustrate how fuzzy semantic self-feedback constructs preference pairs from diverse candidate responses (Appendix D.1), and then show concrete examples of hallucination reduction in the final fine-tuned model (Appendix D.2).

### D.1. Preference Pairs Cases

To better illustrate how RLSF-V constructs training supervision from fuzzy semantic self-feedback, we present several representative preference pairs (generated by LLaVA-v1.5-7B) in Figure 5-Figure 8. For each multimodal input, we show the image, the user question, and two candidate responses sampled from the base model: the *chosen* response $y^{cho}$ and the *rejected* response $y^{rej}$. These two responses are selected from the full candidate set using the POS-aware fuzzy semantics $U_{pos}(\cdot)$ defined in Equation (13), and we display their corresponding hallucination scores alongside the texts.

In each example, hallucinated segments in the rejected response are highlighted, covering typical failure modes such as introducing non-existent objects, fabricating attributes, or making unsupported statements about the scene. In contrast, the

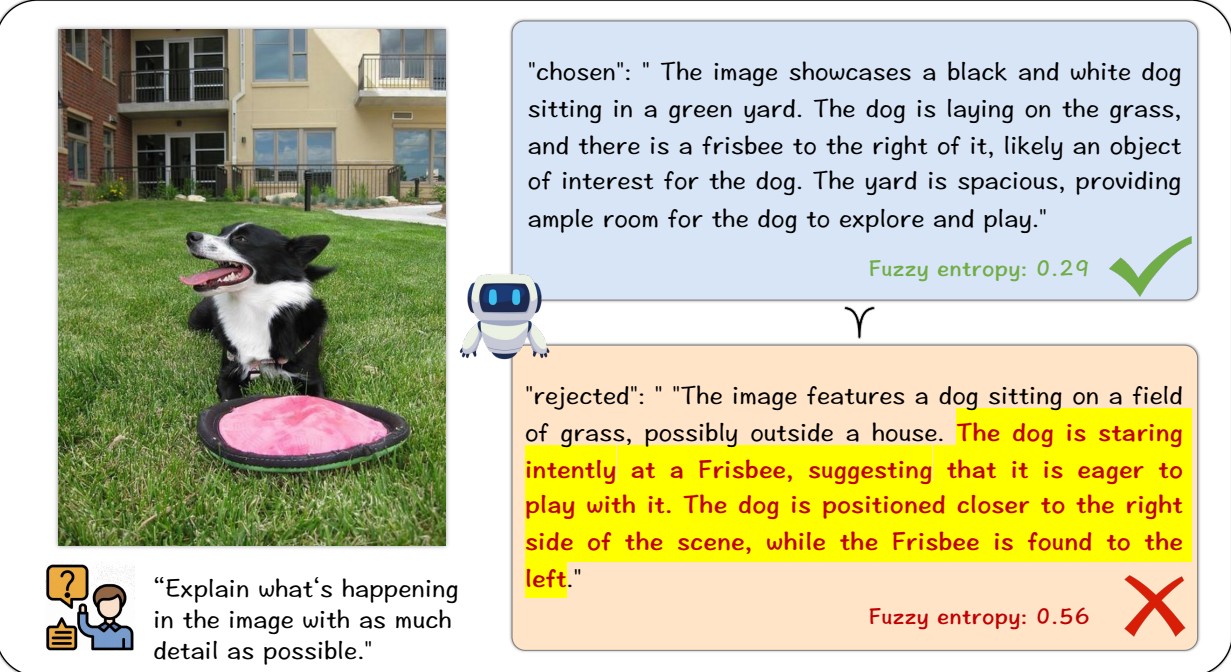

Figure 6. Case study of the fuzzy semantics feedback preference pairs.

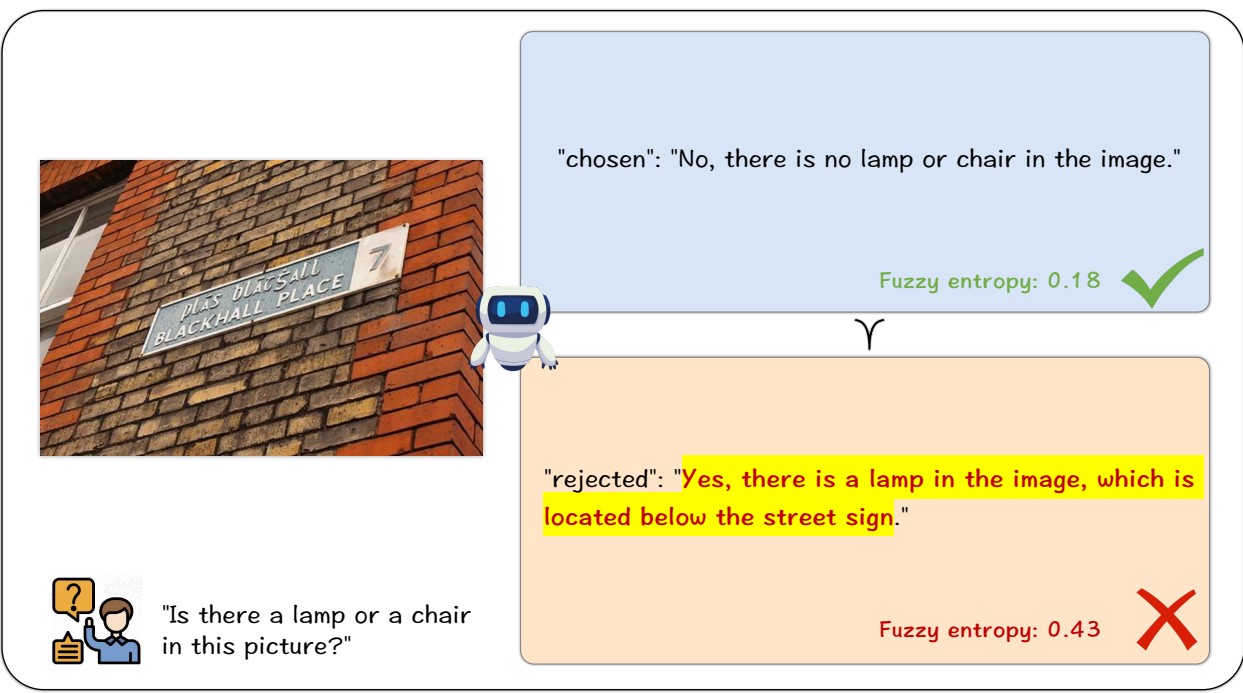

Figure 7. Case study of the fuzzy semantics feedback preference pairs.

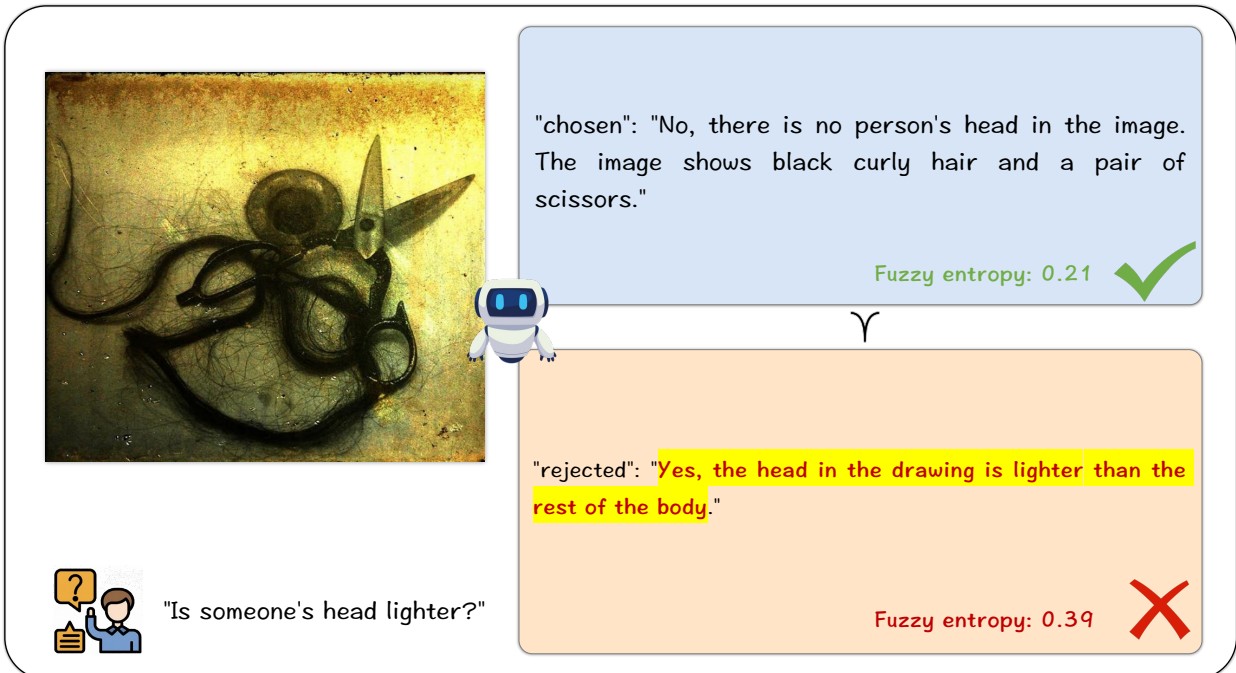

*Figure 8.* Case study of the fuzzy semantics feedback preference pairs.

chosen response is generally more conservative and better aligned with the visual evidence, often avoiding the highlighted hallucinations while still answering the question. We emphasize that we only visualize the final preference pair $(y^{cho}, y^{rej})$ for each prompt rather than all sampled candidates, since the remaining candidates serve only as intermediates for computing $U_{pos}(\cdot)$ and selecting the most and least reliable responses.

Overall, these qualitative examples verify that the self-generated preferences produced by our fuzzy semantic self-feedback align well with human judgments of hallucination severity. Responses with higher POS-aware fuzzy uncertainty tend to contain more highlighted hallucinations and are consistently ranked as rejected, whereas low-uncertainty responses are selected as preferred. This supports the use of $U_{pos}(\cdot)$ as an effective surrogate signal for constructing preference data without external feedback.

### D.2. Hallucination Mitigation Cases

We next examine how the preferences induced by fuzzy semantic self-feedback translate into improved behavior after fine-tuning. Figure 9 compares the outputs of the base LLaVA-v1.5-7B model and LLaVA-v1.5-7B+RLSF-V on four generative question–answering examples. For each multimodal input, we show the image, the question, and both model responses, with hallucinated spans highlighted. Across these cases, the base model frequently invents objects, attributes, or relations that are not supported by the image, whereas RLSF-V either corrects these hallucinations or refrains from overconfident speculation, leading to more faithful and grounded descriptions.

To further illustrate the effect on discriminative judgments, Figure 10 reports five yes-or-no questions together with the answers from the base model and LLaVA-v1.5-7B+RLSF-V. Here, the highlighted regions indicate incorrect or hallucinated statements (*e.g.*, answering "yes" to the presence of an object that does not exist). We observe that RLSF-V substantially reduces such hallucinations and produces a higher proportion of correct yes-or-no decisions, consistent with our quantitative gains on AMBER Discriminative and other benchmarks.

Finally, Figure 11 and Figure 12 qualitatively compare different self-feedback baselines on both generative and yes-or-no examples. For each input, we show the outputs of models trained with fuzzy semantic self-feedback, energy-based self-feedback, evidence-theory self-feedback, and Shannon-entropy-based self-feedback, with hallucinated spans highlighted. While all variants share the same underlying candidate pools and training data, only our fuzzy semantic self-feedback

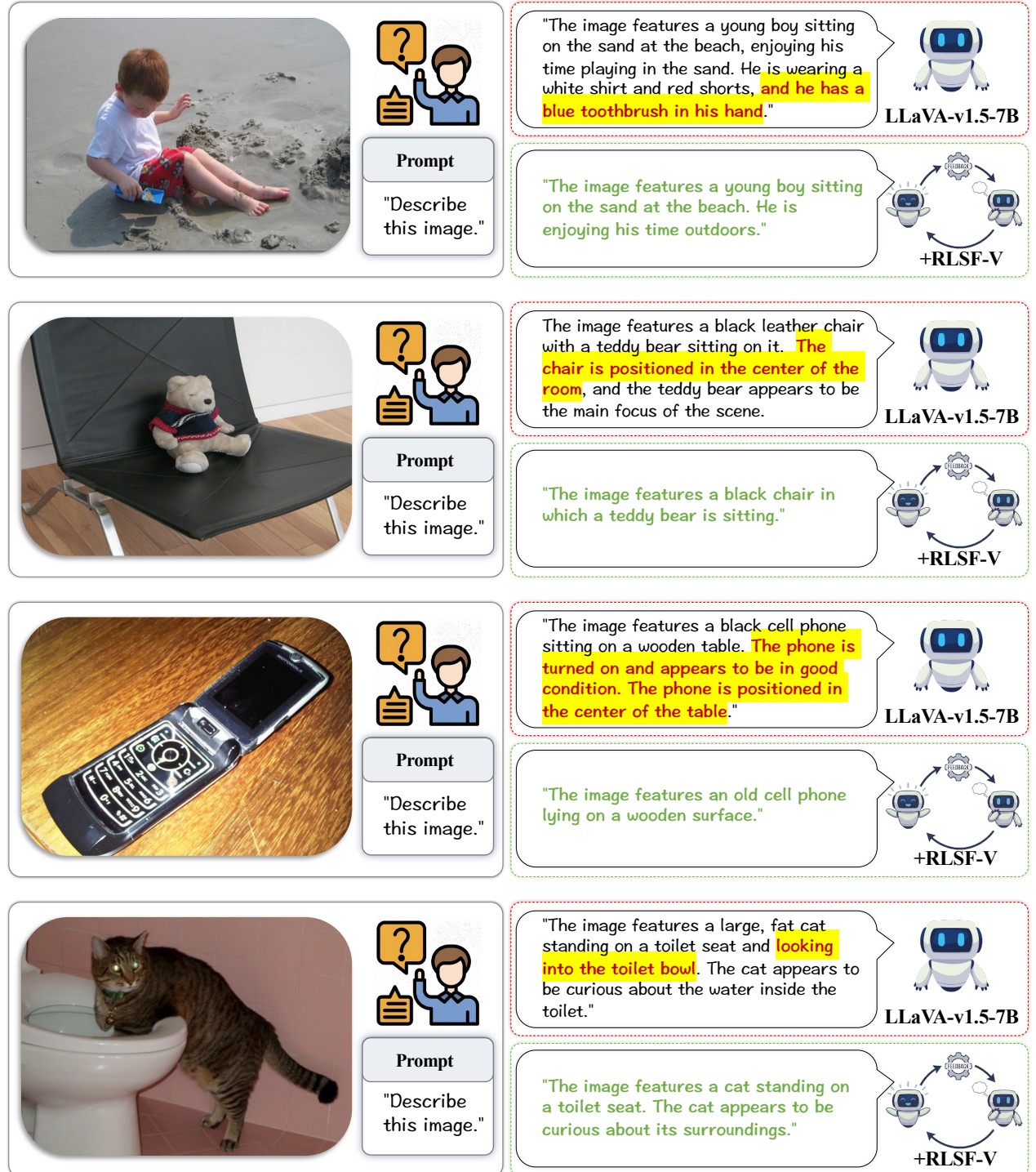

*Figure 9.* Hallucination mitigation cases (generative) between LLaVA-v1.5-7B and our RLSF-V.

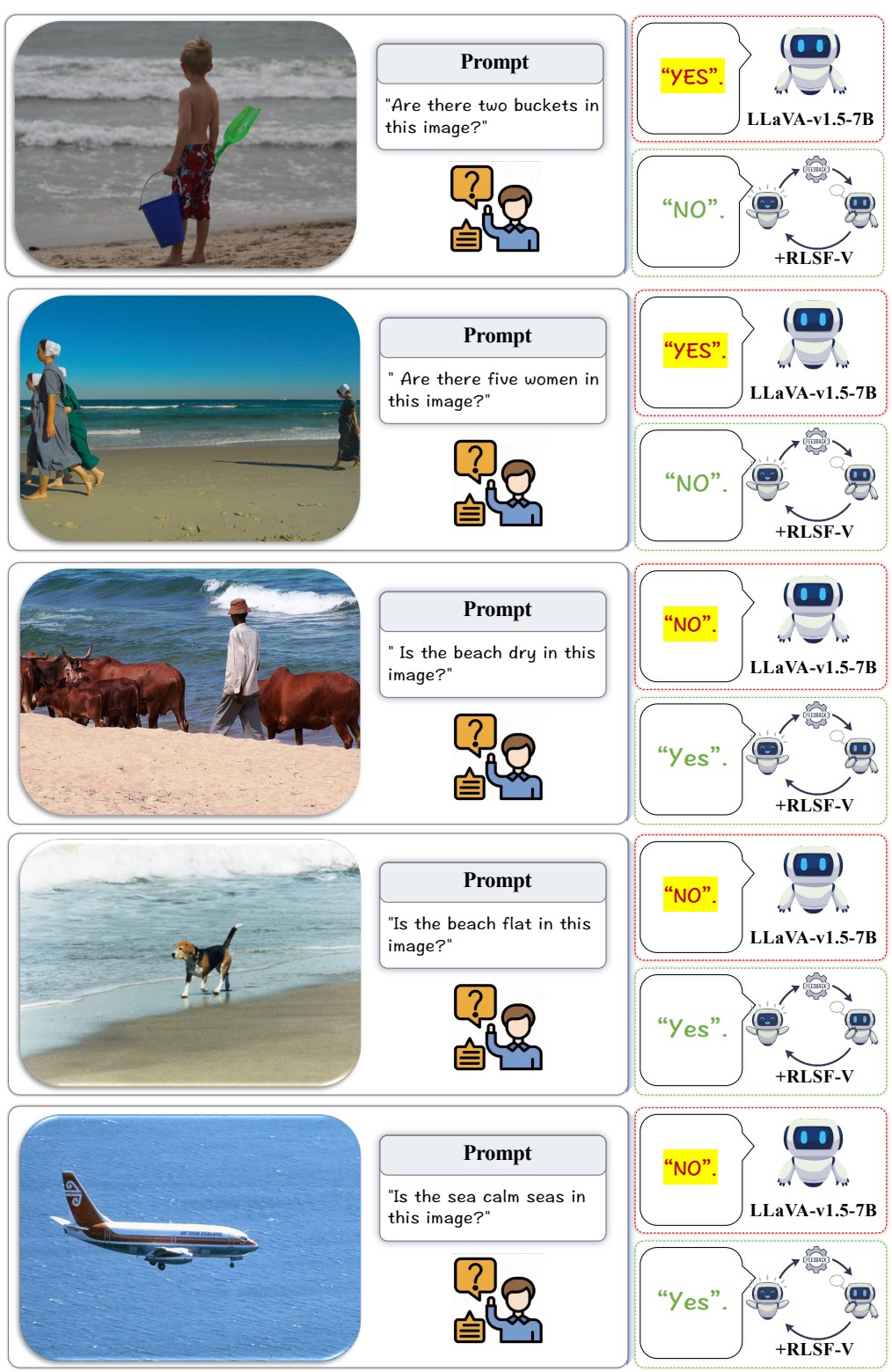

*Figure 10.* Hallucination mitigation cases (discrimative) between LLaVA-v1.5-7B and our RLSF-V.

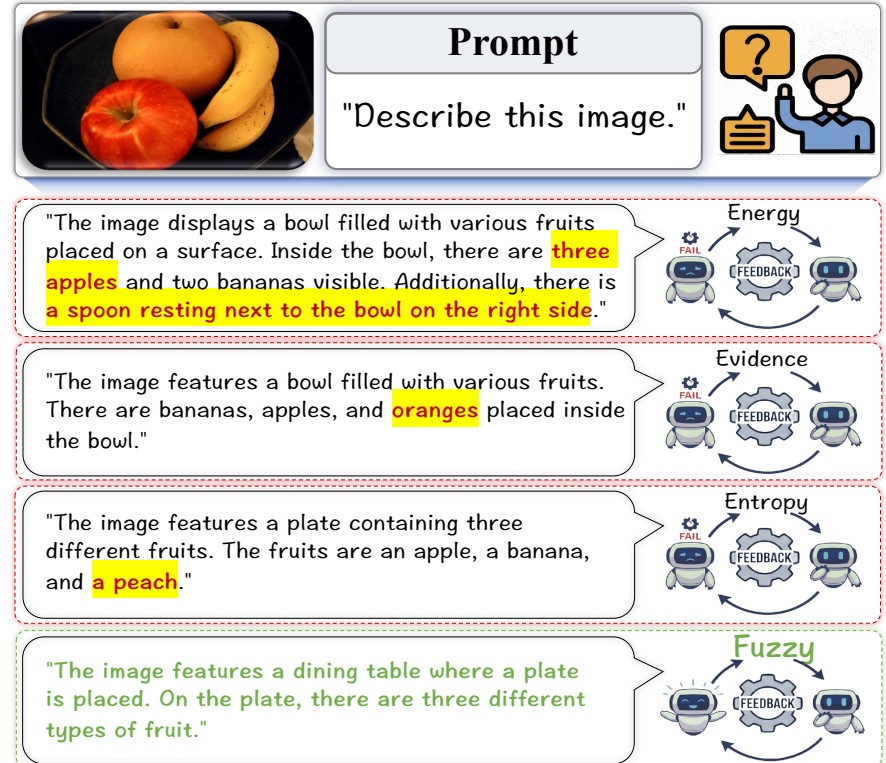

*Figure 11.* Hallucination mitigation cases (generative) between various self-feedback baselines and our local fuzzy semantics feedback.

consistently yields responses with fewer highlighted hallucinations. In contrast, alternative self-feedback signals often lead to models that still produce clearly hallucinated content. These case studies complement the quantitative ablations in the main paper and provide direct evidence that fuzzy semantic self-feedback offers more reliable guidance for preference construction and hallucination mitigation.

## E. Limitations of RLSF-V

Although our study focuses on hallucination mitigation in MLLMs under a specific experimental protocol, several natural extensions remain open for future work. First, we primarily evaluate RLSF-V on a representative set of hallucination-sensitive benchmarks and a widely used multimodal backbone; applying the same framework to a broader range of architectures and task formats (*e.g.*, open-domain dialogue, highly creative captioning, or instruction-heavy reasoning tasks) would further clarify its generality. Second, our fuzzy semantic self-feedback is instantiated with a particular choice of POS-aware token selection and logit-based uncertainty aggregation; exploring alternative linguistic priors, additional forms of internal signals, and more automated ways of tuning these components could yield further gains. Finally, while we compare against strong RLHF- and RLAIF-style baselines, a more comprehensive study that integrates RLSF-V with external feedback or combines it with other training paradigms is an interesting direction, and may help delineate when internal self-feedback alone is sufficient versus when it is best used as a complementary signal.

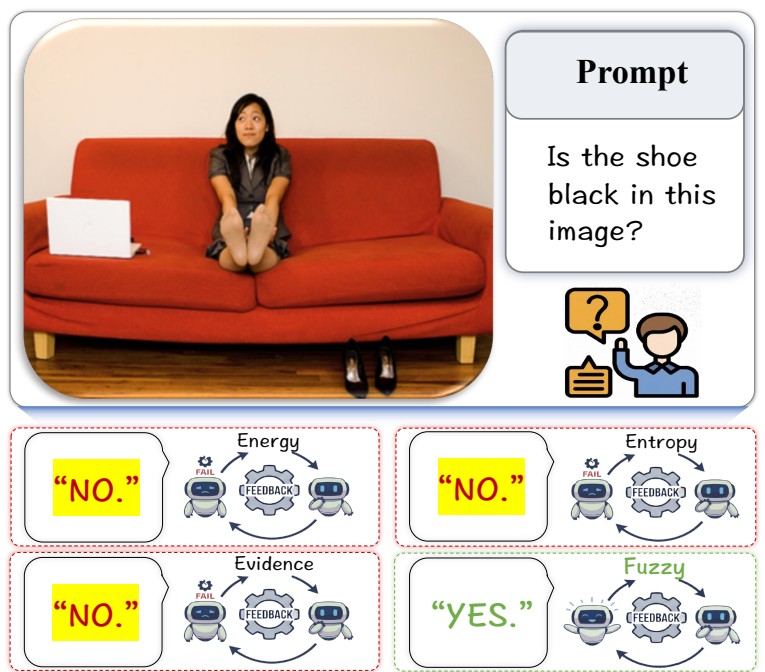

*Figure 12.* Hallucination mitigation cases (discrimative) between various self-feedback baselines and our local fuzzy semantics feedback.

