# OpenReview forum: "RLSF-V: Mitigating Hallucinations in MLLMs via Fuzzy Semantic Self-Feedback"
_ICML.cc/2026/Conference — ICML 2026 regular_

### Official Review · Reviewer_C1EU · 2026-03-11

**Soundness:** 3
**Presentation:** 3
**Significance:** 2
**Originality:** 2
**Overall Recommendation:** 4
**Confidence:** 3

**Summary:**

This paper introduces RLSF-V, a novel self-feedback framework designed to mitigate hallucinations in Multimodal Large Language Models (MLLMs) without relying on costly external supervisors like GPT-4V, which often violate the on-policy assumptions of Direct Preference Optimization (DPO). Instead of using external preference pairs, RLSF-V generates multiple candidate responses and leverages the model's internal logits to quantify token-level uncertainty through fuzzy semantics. To accurately pinpoint hallucinations, it employs POS-aware token selection—focusing only on words likely to carry factual information—and top-R pooling to isolate the most unstable tokens. By automatically treating the response with the lowest aggregated uncertainty as "chosen" and the highest as "rejected," the method enables purely self-supervised DPO fine-tuning. Extensive evaluations on LLaVA-v1.5 across various benchmarks demonstrate that RLSF-V matches or exceeds the performance of strong, externally-supervised baselines, offering a highly scalable and cost-effective solution for MLLM alignment.

**Compliance With Llm Reviewing Policy:**

Affirmed.

**Final Justification:**

The rebuttal addressed my main concerns.

**Key Questions For Authors:**

See the weaknesses. I would like to raise my scores if the authors successfully address my concerns.

**Limitations:**

yes

**Strengths And Weaknesses:**

**Strength:**
1. The topic selection is important.
This article captures an important point: many hallucination mitigation methods, although effective, rely heavily on external large models or manual annotation, resulting in cost, privacy, and scalability issues. At the same time, the authors elevate this further to the DPO's on-policy/off-policy discussion, which makes the problem setting more methodological than simply reducing hallucinations.
2. The method design is relatively complete.
Instead of proposing just a token-level uncertainty metric, RLSF-V systematically embeds it into the entire chain of candidate generation, token scoring, POS-aware filtering and DPO fine-tuning. The overall pipeline is described more coherently in Figure 2 and in the Methodology section.
3. The experiment is relatively sufficient.
The paper also compares a large number of RLHF / RLAIF / DPO class methods, as well as different internal uncertainty signals. Table 1, Table 2, Table 3, and Figure 4 basically cover key elements such as the main result, component ablation, alternative scoring function, hyperparameter analysis, etc.

**Weaknesses:**
1. The author repeatedly emphasizes that the external model correction results in an off-policy preference, so it is not ideal. But from a practical standpoint, many DPO variants don't strictly require strong on-policy data to be effective. The authors should prove at the theoretical level that the off-policy problem here is the main bottleneck.
2. The authors only conducted experiments on LLaVA-based models. First, the model is outdated, and secondly, the generalizability of the method cannot be validated by experimenting only on one model architecture.
3. The authors should provide the performance of the method on more general-purpose multimodal datasets.

---

> ### Author Rebuttal · Authors · 2026-03-31
>
> Thank you for the constructive feedback and for recognizing the importance of reducing reliance on external supervisors. We are encouraged that you find the problem setting important and the overall pipeline relatively complete. We respond to each concern below.
>
> ---
>
> ### **Weakness 1 (on-/off-policy data in DPO)**
>
> We agree that many practical DPO variants can still be effective without strongly on-policy data, and we do not claim otherwise. Our point is narrower: when a preferred response has very low probability under the initial policy, the resulting distribution mismatch can make DPO optimization less effective. **Prior theoretical analysis in OPA-DPO [1] supports this view**, suggesting that if a preferred response has near-zero probability under the initial policy, DPO may struggle to learn it effectively. Intuitively, DPO is better at reweighting plausible in-distribution responses than at learning preferred responses far outside the model’s own generation distribution. **Thus, severe off-policy mismatch is not universally fatal, but it is a meaningful factor that can weaken DPO training**.
>
> This motivates our self-feedback construction. Both the chosen and rejected responses are sampled from the same base MLLM, so the resulting preference pairs stay closer to the reference distribution than externally corrected or rewritten responses. Our claim is therefore not that DPO must be strictly on-policy, but that reducing this mismatch yields a cleaner and more compatible preference construction pipeline for DPO-based hallucination mitigation.
>
> We will revise the manuscript to **remove or weaken statements that describe the off-policy issue as the main bottleneck**.
>
> _[1] Mitigating Hallucinations in Large Vision-Language Models via DPO: On-Policy Data Hold the Key, CVPR 2025_
>
> ---
>
> ### **Weakness 2 (evaluation only on LLaVA)**
>
> To directly address this concern, we additionally evaluate RLSF-V on three non-LLaVA model families: **Qwen2.5-VL**, **Qwen3-VL** and **InternVL3_5**. RLSF-V consistently improved upon the corresponding baseline models across most representative hallucination metrics.:
>
> |Model|AMBER GEN|HallusionBench|Object HalBench|AMBER DIS|
> |---------------------------|----------------------------|------------------|-------------------|---------------|
> ||**CHAIR / HalRate. / Cog.↓**|**All Acc.↑**|**Resp. / Ment.↓**|**Acc. / F1.↑**|
> |InternVL3_5-8B-HF|7.6 / 63.8 / 5.3|57.6|5.5 / 9.4|**86.7 / 89.4**|
> |**InternVL3_5-8B-HF+RLSF-V**|**7.2 / 58.7 / 4.5**|**58.1**|**4.4 / 6.7**|**86.7 / 89.4**|
> |Qwen2.5-VL-3B-Instruct|8.0 / 49.1 / 5.2|53.1|9.7 / 15.7|88.0 / 91.0|
> |**Qwen2.5-VL-3B-Instruct+RLSF-V**|**6.5 / 36.5 / 3.9**|**53.9**|**7.0 / 9.3**|**88.2 / 91.1**|
> |Qwen3-VL-8B-Instruct|8.1 / 62.0 / 3.4|64.0|8.1 / 14.3|89.0 / 91.7|
> |**Qwen3-VL-8B-Instruct+RLSF-V**|**6.7 / 48.1 / 2.3**|**65.5**|**5.5 / 9.7**|**89.5 / 92.1**|
>
> These additional results suggest that the proposed self-feedback mechanism is **not** tied to a single LLaVA architecture, **but can transfer to the tested Qwen/InternVL models well**. We will include these results in the revised version.
>
> ---
>
> ### **Weakness 3 (more general-purpose multimodal datasets)**
>
> To assess broader multimodal capability beyond hallucination benchmarks, we additionally evaluate **Qwen3-VL-8B-Instruct + RLSF-V** on a diverse suite of general-purpose benchmarks, including **reasoning** (MMMU, MMStar), **perception / cognition** (MME), **diagram and real-world QA** (AI2D, RealWorldQA), and **text-rich understanding** (ChartQA, DocVQA, OCRBench). RLSF-V remains broadly competitive on general-purpose benchmarks, improving **most** benchmarks—including **AI2D, ChartQA, DocVQA, MMMU, MMStar, RealWorldQA, and SeedBench**—with only minor trade-offs on **OCRBench and MME**. This suggests that improved hallucination mitigation does not appear to come at a large cost to general capability in these tests.
>
> |Benchmark|Qwen3-VL-8B-Instruct|Qwen3-VL-8B-Instruct + RLSF-V|
> |--------------|-----------|------------------------|
> |MME-Cognition|**641.79**|638.93|
> |MME-Perception|**1720.80**|1715.30|
> |MMMU Eval|52.33|**52.89**|
> |MMStar|62.88|**63.78**|
> |AI2D|83.65|**83.71**|
> |RealWorldQA|69.41|**70.20**|
> |ChartQA|85.36|**85.52**|
> |OCRBench|**83.00**|82.40|
> |SeedBench|75.07|**75.15**|
> |DocVQA|95.61|**95.95**|
>
> ---
>
> We sincerely hope these clarifications and the new experiments help address your concerns.

---

> > ### Author Rebuttal · Reviewer_C1EU · 2026-04-05
> >
> > Thanks for the rebuttal and additional results, which successfully addressed my concerns.

---

> > > ### Author Response · Authors · 2026-04-05
> > >
> > > Thank you again for your thoughtful review and for raising your score after the rebuttal!
> > >
> > > We especially appreciate your suggestions on clarifying the paper’s positioning and strengthening the broader evaluation, which were very helpful in improving the paper.

---

### Official Review · Reviewer_fgg4 · 2026-03-11

**Soundness:** 4
**Presentation:** 3
**Significance:** 4
**Originality:** 4
**Overall Recommendation:** 5
**Confidence:** 5

**Summary:**

This paper proposed RLSF-V, a fully on-policy self-feedback framework for mitigating hallucination in MLLM without human or external model feedback. The self-feedback is driven by the proposed local fuzzy semantics, which automatically constructs a preference dataset for DPO. The motivation and design are strong, and extensive experiments on multiple hallucination benchmarks and general capability benchmarks, as well as ablation studies on local fuzzy semantics, support the effectiveness of the proposed ideas and methods.

**Compliance With Llm Reviewing Policy:**

Affirmed.

**Final Justification:**

The author addresses my concerns with extra experiments.  Therefore, I maintain my decision.

**Key Questions For Authors:**

1.  In Definition 3.1, is local fuzzy semantics performed on logits across the entire vocabulary or on a truncated vocabulary? Are other uncertainty methods consistent?

2.  In evidence-based methods, why use AU\*EU? What about using only one of them?

**Limitations:**

yes.

**Strengths And Weaknesses:**

Strengths:
1.  The proposed self-feedback is highly meaningful and valuable, as it eliminates the need for manual or expensive external model feedback. This approach has not yet been addressed in MLLM hallucination mitigation.
2.  RLSF-V achieves competitive results on multiple hallucination benchmarks with a relatively small dataset, and also performs well on general benchmarks.
3.  The construction and formulation of local fuzzy semantics are solid, incorporating other classic uncertainty measures, and the ablation is good.
4.  The paper is well-written, highlighting key points and offering novel insights for the design of other hallucination mitigation methods.

Weaknesses:
1.  Only results on LLaVA are shown in the manuscript. Results on other model families, e.g. Qwen, could further improve the scalability.
2.  The implementation details of spacy are not clear enough. What version of the model was used, and does a stronger or weaker model affect the results?
3.  Methods based on steering vectors as hallucination mitigation methods should also be included in the discussion.
4.  Performance for some closed-source models could be added to the tables.

---

> ### Author Rebuttal · Authors · 2026-03-31
>
> We thank the reviewer for the positive assessment and for recognizing the value of our self-feedback framework, local fuzzy semantics, and the empirical results.
>
> ---
>
> ### **Weakness 1 (generalization to other model families)**
>
> We agree that cross-family validation is important. We therefore additionally ran RLSF-V on three non-LLaVA model families: **Qwen2.5-VL**, **Qwen3-VL** and **InternVL3_5**, and observed improvements on most reported hallucination metrics across the tested backbones.
>
> |Model|AMBER GEN|HallusionBench|Object HalBench|AMBER DIS|
> |---------------------------|----------------------------|------------------|-------------------|---------------|
> ||**CHAIR / HalRate. / Cog.↓**|**All Acc.↑**|**Resp. / Ment.↓**|**Acc. / F1.↑**|
> |InternVL3_5-8B-HF|7.6 / 63.8 / 5.3|57.6|5.5 / 9.4|**86.7 / 89.4**|
> |**InternVL3_5-8B-HF + RLSF-V**|**7.2 / 58.7 / 4.5**|**58.1**|**4.4 / 6.7**|**86.7 / 89.4**|
> |Qwen2.5-VL-3B-Instruct|8.0 / 49.1 / 5.2|53.1|9.7 / 15.7|88.0 / 91.0|
> |**Qwen2.5-VL-3B-Instruct + RLSF-V**|**6.5 / 36.5 / 3.9**|**53.9**|**7.0 / 9.3**|**88.2 / 91.1**|
> |Qwen3-VL-8B-Instruct|8.1 / 62.0 / 3.4|64.0|8.1 / 14.3|89.0 / 91.7|
> |**Qwen3-VL-8B-Instruct + RLSF-V**|**6.7 / 48.1 / 2.3**|**65.5**|**5.5 / 9.7**|**89.5 / 92.1**|
>
> These results suggest that RLSF-V is not specific to the LLaVA family and generalizes to other MLLM backbones.
>
> ---
>
> ### **Weakness 2 (POS-tagger implementation details)**
>
> Thank you for pointing this out. We used **spaCy 3.8.0** with **`en_core_web_sm`**. The POS tagger is only used to build a coarse mask over content-bearing tokens, while the self-feedback signal still comes from the model’s internal logits. We additionally tested other spaCy English pipelines on LLaVA-v1.5-7B and found very similar results:
>
> |spaCy size|AMBER GEN|HallusionBench|Time(s)|
> |----------|-------------|------------------|-----------|
> ||**CHAIR / HalRate. / Cog.↓**|**All Acc.↑**||
> |sm|2.4 / 5.1 / 0.3|47.4|11|
> |md|2.3 / 5.0 / 0.3|47.4|12|
> |lg|2.5 / 5.3 / 0.3|47.3|25|
> |trf|2.5 / 5.2 / 0.3|47.8|399|
>
> Since we only use coarse POS tagging, stronger pipelines bring little performance gain but much higher runtime. We therefore use **`en_core_web_sm`** as an efficient default. The complete implementation details are provided in **Appendix B.1**.
>
> ---
>
> ### **Weakness 3 & 4 (relation to steering-vector methods and closed-source comparisons)**
>
> We appreciate these suggestions. We will add a discussion of steering-vector / activation-intervention methods in the Related Work section, and we will also add closed-source MLLM results on the hallucination benchmarks to the tables in the revision where feasible.
>
> ---
>
> ### **Question 1 (clarification on Definition 3.1 and baseline consistency)**
>
> Thank you for this important question. Formally, local fuzzy semantics is defined over the set of tokens with finite logits:
>
> $\mathcal{S}\_t = \\{ v \in \mathcal{V} \mid z\_{t,v} > -\infty \\}$.
>
> In practice, since we use **top-k sampling** during generation, we cache the same **top-k logits** with **k = 50**. Thus, $\mathcal{S}_t$ corresponds to this retained **top-k candidate set** at each step. For consistency and fair comparison, all uncertainty measures in our experiments are computed on this same cached candidate set. We will clarify this implementation detail in the revision.
>
> ---
>
> ### **Question 2 (why use AU * EU)**
>
> We use AU * EU because this is the evidential uncertainty formulation adopted from [1] and detailed in our Appendix A. Here, EU is high when the total evidence is small, and AU is high when the evidence is dispersed across competing candidates. Their product therefore highlights positions that are both weakly supported and ambiguous. We additionally tested AU-only and EU-only variants on AMBER GEN with LLaVA-v1.5-13B:
>
> |**EU**|**AU**|**CHAIR↓**|**HalRate.↓**|**Cog.↓**|
> |---|---|---:|---:|---:|
> |✔||6.8|27.4|2.9|
> ||✔|2.4|5.0|0.3|
> |✔|✔|**1.7**|**3.2**|**0.3**|
>
> These results suggest that AU captures useful local ambiguity, while combining it with EU further improves the evidence-based self-feedback.
>
> _[1] Estimating LLM Uncertainty with Evidence, arXiv 2025_
>
> ---
>
> We thank the reviewer again for the constructive suggestions, which helped improve both the clarity and the scope of the paper.

---

> > ### Author Rebuttal · Reviewer_fgg4 · 2026-04-02
> >
> > The author address my concerns with extra experiments. I will keep my positive score.

---

> > > ### Author Response · Authors · 2026-04-05
> > >
> > > Thank you again for your encouraging review and for reading our rebuttal so carefully. We sincerely appreciate your recognition that the additional experiments addressed your concerns, and your feedback helped us further improve the paper’s presentation.

---

### Official Review · Reviewer_58iY · 2026-03-12

**Soundness:** 3
**Presentation:** 3
**Significance:** 3
**Originality:** 3
**Overall Recommendation:** 5
**Confidence:** 4

**Summary:**

The paper propose a novel preference learning framework that leverage logits as self-feedback to construct preference pairs for training. Specifically, the author propose a local fuzzy score to quantify token-level uncertainty on tokens of certain POS tags such as nouns. It further adopt a top-R pooling strategy and then generate high-quality pairs. Experimental results on LLaVA 1.5 7B and 13B models demonstrate the effectiveness of the proposed method.

**Compliance With Llm Reviewing Policy:**

Affirmed.

**Final Justification:**

The rebuttal response solved my concerns and I raise my score to 5.

**Key Questions For Authors:**

See strength and weakness.

**Limitations:**

yes

**Strengths And Weaknesses:**

Strength
1. The proposed fuzzy score as a self-feedback signal is novel and intuitive.
2. The pos-tag-based filtering and top-R pooling strategy is effective and verified by detail ablations.
3. The empirical performance improvement is significant on diverse hallucination-related benchmarks.


Weakness
1. Core assumption analysis. The method assumes internal uncertainty positively correlates with higher hallucination risk. Though intuitively it could often be true, but clearly not alway true. Models could confidently make mistakes. The paper should give more analysis on this failure mode.
2. Risk of over-claim. The paper discuss about the on-policy problem in the introduces as one of the core motivation of the proposed method. However many existing works like TPO and RLAIF-V already adopt on-policy data during training.
3. Evaluation. Since the method explicitly encourage greater certainty, the author may have to also evaluate the model output diversity and performance on more general benchmarks.
4. Experimental setting. All experiments are conducted on LLaVA v1.5 7B/13B which are relatively dated. The author could improve more recent baselines like Qwen2.5-VL, Qwen3.5 to demonstrate the generalization and effectiveness of the proposed method.

---

> ### Author Rebuttal · Authors · 2026-03-31
>
> We thank the reviewer for the constructive feedback and the positive assessment of the novelty and empirical effectiveness of our method. We address the concerns below.
>
> ---
>
> ### **Weakness 1 (core assumption/failure mode)**
>
> We agree that internal uncertainty is not always aligned with hallucination risk, and that models can indeed make confident mistakes. Our method does not require uncertainty to be a universally correct factuality estimator; rather, it only needs a useful **local ranking signal** among responses sampled from the same model for the same prompt. To verify this, we additionally performed a direct ranking-quality evaluation on AMBER (**detailed protocol is provided in our response to Reviewer vJ83, Weakness 1**). Using the official AMBER hallucination labels as ground truth, our local fuzzy semantics achieves **71.6 AUROC and 77.7 pairwise accuracy**, which supports that it is useful for candidate-level ranking on AMBER, even if uncertainty is not perfectly equivalent to factual correctness.
>
> At the same time, we agree that the manuscript does not sufficiently discuss failure modes. One representative case is **fine-grained relational reasoning**, where the model may still make a confident but visually misgrounded prediction on subtle relations such as left/right or in front of/behind, **especially when ADP tokens are included in our POS-tagging strategy** (as also reflected in the third observation of Sec. 4.2 in the manuscript). We will revise the manuscript to clarify this and add the corresponding discussion.
>
> ---
>
> ### **Weakness 2 (over-claim issue)**
> We fully agree that our current wording is too strong. Our intention was not to claim that this is a main contribution of our work, but rather to motivate why a fully self-contained self-feedback pipeline is appealing. We will revise the manuscript to avoid over-emphasizing the on-policy aspect, and instead focus the contribution more clearly on the proposed self-feedback mechanism.
>
> ---
>
> ### **Weakness 3 & 4 (diversity / general benchmarks / more recent backbones)**
>
> We agree that hallucination reduction should not simply make the model overly conservative, and that evaluation only on LLaVA-v1.5 is limited. To address both concerns, we added experiments on **more recent MLLM backbones** and examined both hallucination-related and diversity / informativeness-related metrics. Here, **AMBER and Object HalBench** are the same benchmarks used in the manuscript, while **MMHal-Bench and CHAIR** are **two additional** hallucination benchmarks that we introduced specifically to further assess whether hallucination mitigation comes at the cost of response diversity / informativeness. For each, the first row reports the **diversity / informativeness metric**, while the second reports the **hallucination metric**.
>
> |Benchmark|Metric|Qwen2.5-VL-3B-Instruct|Qwen2.5-VL-3B-Instruct + RLSF-V|Qwen3-VL-8B-Instruct|Qwen3-VL-8B-Instruct + RLSF-V|
> |---------------|----------------------------|------------:|-------------------:|-------------:|-----------------:|
> |AMBER GEN|Cover↑|**69.3**|66.2|**74.5**|71.7|
> ||CHAIR↓ / HalRate.↓ / Cog.↓|8.0 / 49.1 / 5.2|**6.5 / 36.5 / 3.9**|8.1 / 62.0 / 3.4|**6.7 / 48.1 / 2.3**|
> |Object HalBench|Recall↑|**52.9**|51.8|52.2|**52.4**|
> ||Resp.↓ / Ment.↓|9.7 / 15.7|**7.0 / 9.3**|8.1 / 14.3|**5.5 / 9.7**|
> |MMHal-Bench|Informativeness↑ / Score↑|59.38 / 3.78|**60.42 / 3.91**|**83.33** / 4.47|81.25 / **4.69**|
> ||HalRate.↓|33.3|**30.2**|34.4|**25.0**|
> |CHAIR|Recall↑|**70.6**|66.0|**74.9**|72.3|
> ||CHAIRs↓ / CHAIRi↓|36.0 / 9.6|**32.0 / 9.2**|48.2 / 10.5|**43.6 / 8.7**|
>
> These results suggest that **RLSF-V generalizes beyond LLaVA-v1.5**. While some coverage/recall drop slightly, we do not observe obvious collapse into generic responses on these metrics. For example, on Qwen2.5-VL-3B-Instruct, MMHal-Bench **Informativeness** improves from **59.38 to 60.42** while hallucination metrics are consistently reduced; on Qwen3-VL-8B-Instruct, MMHal-Bench **Score** improves from **4.47 to 4.69** and **HalRate** drops from **34.4 to 25.0**.
>
> Due to the rebuttal character limit, we provide the full results on broader **general-purpose multimodal benchmarks** in our response to **Reviewer C1EU, Weakness 3**. Those results show that RLSF-V remains broadly competitive on benchmarks such as AI2D, ChartQA, DocVQA, MMMU, MMStar, RealWorldQA, and SeedBench, supporting that the hallucination reduction is unlikely to be explained solely by a major collapse in general capability.
>
> For additional results beyond LLaVA-v1.5, including experiments on **InternVL3_5-8B-HF**, please also refer to our response to **Reviewer C1EU, Weakness 2** due to the same space constraint.
>
> Together, these results further support that RLSF-V transfers to more recent backbones while largely preserving informativeness on the tested settings.
>
> ---
>
> We sincerely appreciate the reviewer’s thoughtful comments and believe they will help strengthen the final version of the manuscript.

---

> > ### Author Rebuttal · Reviewer_58iY · 2026-04-03
> >
> > The response is comprehensive and clear. The authors have addressed my concerns, and I raise my score to 5.

---

> > > ### Author Response · Authors · 2026-04-05
> > >
> > > Thank you again for your thoughtful review and for raising your score after the rebuttal. We truly appreciate your recognition that our response was comprehensive and clear, and your suggestions were very helpful in strengthening the paper.

---

### Official Review · Reviewer_vJ83 · 2026-03-13

**Soundness:** 3
**Presentation:** 3
**Significance:** 3
**Originality:** 3
**Overall Recommendation:** 3
**Confidence:** 4

**Summary:**

The paper proposes RLSF-V, a self-feedback pipeline for hallucination mitigation in MLLMs. For each image-text prompt, it samples 10 candidate responses from LLaVA-v1.5, scores each using a new token-level “local fuzzy semantics” uncertainty derived from the model’s own logits plus POS-aware top-R token pooling, then forms DPO pairs by taking the lowest- vs highest-uncertainty responses. The main claim is that this fully self-generated, “on-policy” preference data can outperform RLHF/RLAIF baselines on hallucination benchmarks without any external models or human labels.

**Compliance With Llm Reviewing Policy:**

Affirmed.

**Final Justification:**

The authors have mostly addressed my concerns.

**Key Questions For Authors:**

1. Do the authors consider directly evaluating candidate-ranking quality of fuzzy vs entropy/evidence on a labeled set of sampled responses (e.g., pairwise accuracy or AUC for hallucination severity), rather than only downstream DPO results in Table 3?
2. The “strictly on-policy” claim seems to hold only for data collected from the initial base model. After DPO updates, why is this still on-policy in the sense required by your argument against external-feedback methods?
3. In Table 1, how much of RLSF-V’s gain comes from self-feedback versus simply using best-of-10 / worst-of-10 sampled responses? What happens with a random chosen/rejected pair or a length-normalized log-probability ranking under the same candidate pool?

**Limitations:**

The limitations section misses: (1) the risk that low-uncertainty responses are merely shorter/more conservative rather than more factual, which matters because Eq. 14 always picks min/max uncertainty candidates; (2) the lack of evidence that the method transfers beyond LLaVA-v1.5 7B/13B despite claims of applicability to “diverse MLLMs”

**Strengths And Weaknesses:**

Strength:
- The paper does include targeted ablations: Table 2 isolates POS-aware selection and top-R pooling, and Table 3 swaps the uncertainty score while keeping the rest of the pipeline fixed.
- The method is clearly specified, with equations for the score and Algorithm 1 for data construction.
- Avoiding external evaluators is a meaningful systems contribution if performance holds.

Weaknesses
- The paper claims the fuzzy score is a better hallucination assessor, but the evidence only shows better *downstream DPO results* in Table 3. There is no direct evaluation of whether \(U_{pos}\) actually ranks hallucinated responses better than entropy/evidence on labeled candidate sets, so the central mechanism is not validated independently of DPO optimization noise.
- The paper claims “strictly on-policy” preference construction, but Section 3.4 builds pairs once from samples of the base model and then optimizes DPO against a changing policy. That is at best on-policy w.r.t. the initial reference model, not the updated policy; the theoretical claim is overstated and not tested.
- Proposition 3.2 proves invariance to positive affine logit transforms for the min-max normalization in Eq. 4, but this is a weak justification for hallucination detection. Table 3 does not isolate whether gains come from scale invariance, from top-R pooling, or simply from selecting conservative low-entropy responses.
- The improvement over baselines in Table 1 is hard to interpret because the training data are not matched. RLSF-V uses 10k prompts sampled from RLAIF-V, while baselines use anywhere from 4.8k to 122k preference pairs from different sources and curation pipelines; this does not support causal claims about the scoring method itself.
- Figure 1/abstract emphasize “over 50% relative reduction in HalRate on AMBER compared to the GPT-4V feedback baselines,” but Table 1 shows the strongest GPT-4V baseline OPA-DPO at 12.1 vs 5.1 (7B), while SENTINEL at 14.8 is not GPT-4V-based. The headline mixes baselines and is rhetorically stronger than the table warrants.
- Figure 3 discusses relation failures, but only for 13B and only against two baselines. The paper does not show whether the same pattern holds for 7B or whether POS tag choice (including ADP) is the culprit.

---

> ### Author Rebuttal · Authors · 2026-03-31
>
> We thank the reviewer for the constructive feedback. Below we address each concern and summarize our additional analyses.
>
> ---
>
> ### **Weakness 1 & Question 1 (direct ranking quality)**
> We added a direct ranking-quality evaluation on AMBER under **the same decoding setup** as the manuscript. We sampled candidate responses, labeled them with the official AMBER hallucination protocol, and evaluated three feedback signals using AUROC and Pairwise Acc:
>
> |Feedback|AUROC|Pairwise Acc|
> |---|---:|---:|
> |Fuzzy|**71.6**|**77.7**|
> |Entropy|70.0|75.1|
> |Evidence|66.5|69.1|
>
> These results suggest that our method (Fuzzy) not only performs better on downstream task, but also serves as a better ranking signal. We will include this in the revision.
>
> ---
>
> ### **Weakness 2 & Question 2 (the on-policy claim)**
> We fully agree that ``strictly/fully on-policy'' was too strong. Our preference data are self-generated from the initial reference model, so they are on-policy only w.r.t the data-collection model, not the continuously updated policy after DPO. We will revise the wording accordingly.
>
> ---
>
> ### **Weakness 3 & Question 3 (what causes the gain)**
> We agree that Proposition 3.2 alone is insufficient, so we added two additional analyses.
>
> First, we applied positive affine transformations to cached logits and re-evaluated ranking quality. Our method (Fuzzy) remains unchanged, while entropy and evidence degrade as the scale increases:
>
> |Feedback|a=1.0,b=1.0|a=2.0,b=1.0|a=4.0,b=1.0|a=8.0,b=1.0|a=16.0,b=1.0|
> |---|---|---|---|---|---|
> |Fuzzy|**71.6 / 77.7**|**71.6 / 77.7**|**71.6 / 77.7**|**71.6 / 77.7**|**71.6 / 77.7**|
> |Entropy|70.0 / 75.1|70.6 / 74.1|69.5 / 71.7|66.9 / 68.0|63.4 / 64.2|
> |Evidence|66.6 / 69.1|65.6 / 67.9|64.6 / 66.3|64.3 / 66.2|64.2 / 66.0|
>
> Each entry is **AUROC/Pairwise Accuracy**.
>
> Second, we compared against random chosen/rejected pairing and length-normalized log-probability ranking. These controls show the gains are not due to arbitrary pairing or solely by a preference for shorter/higher-probability responses:
>
> |Model|Feedback|AMBER GEN|HallusionBench|Object HalBench|AMBER DIS|
> |---|---|---|---|---|---|
> |||**CHAIR / HalRate. / Cog.↓**|**All Acc.↑**|**Resp. / Ment.↓**|**Acc. / F1.↑**|
> |LLaVA-v1.5-7B|Random chosen/rejected|8.0 / 37.4 / 4.2|42.6|26.7 / 51.0|72.0 / 74.8|
> ||Length-normalized log-probability|8.0 / 31.8 / 1.7|41.3|13.0 / 20.7|77.6 / 81.8|
> ||RLSF-V|**2.4 / 5.1 / 0.3**|**47.4**|**2.9 / 4.0**|**78.8 / 84.0**|
> |Qwen3-VL-8B|Random chosen/rejected|8.0 / 60.9 / 3.7|62.4|7.7 / 13.7|89.4 / 91.9|
> ||Length-normalized log-probability|6.7 / 57.6 / 2.9|62.4|7.2 / 12.3|**89.6** / 92.0|
> ||RLSF-V|**6.7 / 48.1 / 2.3**|**65.5**|**5.5 / 9.7**|89.5 / **92.1**|
>
> ---
>
> ### **Weakness 4 (training data in Tab. 1)**
> We agree that Tab. 1 is an end-to-end comparison rather than a perfectly matched isolation of the scoring function. We will revise the wording to avoid overly strong causal claims. Still, our method uses only 10k self-generated preference pairs, without human feedback, external evaluators, or response rewriting, while achieving competitive results.
>
> ---
>
> ### **Weakness 5 (headline wording)**
> We agree that some headline wording was too strong. We will weaken these claims and make the presentation more precise in the revision.
>
> ---
>
> ### **Weakness 6 (relation analysis)**
> We further analyzed relation hallucinations and found the same pattern on both 7B and 13B: **RLSF-V is better overall, but relation remains weaker than LLaVA-RLHF.**
>
> |ModelSize|Method|Overall F1|Relation F1|
> |---|---|---:|---:|
> |13B|RLSF-V|**87.6**|63.1|
> ||LLaVA-RLHF|85.2|**78.1**|
> |7B|RLSF-V|**84.0**|60.4|
> ||LLaVA-RLHF|83.2|**62.2**|
>
> **Thus, this weakness is not limited to 13B**, though the gap is smaller on 7B.
>
> Figure 3 included only LLaVA-RLHF and SENTINEL because adding more baselines would over-clutter the radar plot; they were chosen as representative RLHF- & RLAIF-style methods. The revised version will include complete per-type results.
>
> **We also tested 13B model after removing ADP from POS filtering:**
>
> |Setting|Relation Acc|Relation F1|Overall Acc|Overall F1|
> |---|---:|---:|---:|---:|
> |RLSF-V|76.0|63.1|84.8|87.6|
> |w/o ADP|**78.0**|**75.3**|**86.9**|**90.0**|
>
> Removing ADP **improves the metric, but hurts generative metrics:**
>
> |Setting|CHAIR/HalRate./Cog.↓|Resp./Ment.↓|
> |---|---:|---:|
> |RLSF-V|**2.0/4.4/0.3**|**1.2/1.7**|
> |w/o ADP|2.3/5.4/0.4|2.3/3.3|
>
> So ADP is a trade-off factor: removing it may help relation discrimination, but weakens overall generative hallucination mitigation.
>
> ---
>
> ### **Limitation**
> We agree and will revise the limitations accordingly. We will also clarify that lower-uncertainty responses may be more conservative rather than more factual. Due to space limits, please see our responses to **Reviewer 58iY (Weaknesses 3 & 4)** for diversity analysis, and **Reviewer C1EU (Weakness 2)** for results on Qwen/InternVL.
>
> ---
>
> We sincerely hope these clarifications and additional results address the reviewer’s concerns.

---

> > ### Author Rebuttal · Reviewer_vJ83 · 2026-04-03
> >
> > The authors have mostly addressed my concerns.

---

> > > ### Author Response · Authors · 2026-04-04
> > >
> > > Thank you again for your time and for carefully reading our rebuttal!
> > >
> > > Since this is our last permitted follow-up, we just wanted to briefly note that, beyond the direct ranking-quality analysis and the random / length-normalized controls, we also added **cross-family results on Qwen2.5-VL, Qwen3-VL, and InternVL3_5**, where RLSF-V consistently improves representative hallucination metrics:
> > >
> > > | Model family           | AMBER HalRate.↓ | HallusionBench Acc.↑ | Object HalBench Ment.↓ |  AMBER DIS F1.↑ |
> > > | ---------------------- | --------------: | -------------------: | ---------------------: | --------------: |
> > > | InternVL3_5-8B-HF      | 63.8 → **58.7** |      57.6 → **58.1** |          9.4 → **6.7** | 89.4 → **89.4** |
> > > | Qwen2.5-VL-3B-Instruct | 49.1 → **36.5** |      53.1 → **53.9** |         15.7 → **9.3** | 91.0 → **91.1** |
> > > | Qwen3-VL-8B-Instruct   | 62.0 → **48.1** |      64.0 → **65.5** |         14.3 → **9.7** | 91.7 → **92.1** |
> > >
> > > We further added **broader general-purpose multimodal evaluations** (including **MMMU, MMStar, AI2D, RealWorldQA, ChartQA, OCRBench, SeedBench, DocVQA, and MME**). On **Qwen3-VL-8B**, RLSF-V improves **7/10** reported metrics while remaining close on the others, suggesting that the gains are **not simply due to collapsing into generic or overly conservative responses**.
> > >
> > > Because these additions directly address the transferability and broader-capability concerns, we would be very grateful if you could kindly reconsider whether the revised paper is now closer to a **positive recommendation**.
> > >
> > > In any case, thank you again for the thoughtful review and helpful suggestions — they genuinely improved the paper.

---

### Decision · Program_Chairs · 2026-04-30

**Decision:**

Accept (regular)

**Comment:**

After reading the paper and the rebuttal. The AC agrees with the reviewers on the strong merits of the proposed self-feedback framework, including its novel approach to mitigating hallucinations without relying on expensive external models (R1, R2, R3, R4), the solid design of the local fuzzy semantics (R2, R3), and the extensive supplementary experiments provided during the rebuttal that demonstrate strong cross-model generalization (R2, R3, R4). However, the AC also strongly shares the reviewers' concerns regarding multiple instances of over-claiming in the original manuscript, specifically the overstated "strictly on-policy" claims (R1, R2), the framing of off-policy issues as the primary bottleneck for DPO (R4), and the exaggerated headline phrasing regarding causal improvements (R1). Since the authors have explicitly acknowledged these presentational flaws in the rebuttal and the core methodology remains technically sound and highly impactful, the AC tends to accept this paper. The authors should tone down these over-claims and incorporate the more precise, nuanced wording promised in the rebuttal into the final camera-ready version.